# Control of mitochondrial dynamics by the metabolic regulator dPGC1 limits Yorkie-induced oncogenic growth in *Drosophila*

Wei Qi Guinevere Sew[1]☉, Maria Molano-Fernández[1]☉, Zhiquan Li[2], Artim Lange[1], Nahia Pérez de Ciriza[1], Lene Juel Rasmussen[2], Héctor Herranz[1]*

1 Department of Cellular and Molecular Medicine, University of Copenhagen, Copenhagen, Denmark,
2 Centre for Healthy Aging, Department of Cellular and Molecular Medicine, University of Copenhagen, Copenhagen, Denmark

☉ These authors contributed equally to this work.
* hherranz@sund.ku.dk (HH)

## Abstract

Mitochondrial function and dynamics are essential for maintaining cellular homeostasis and overall health. Disruptions in these processes can contribute to various diseases, including cancer. The Hippo signaling pathway, a key regulator of tissue growth, plays a central role in cancer through its main effector YAP, known as Yki in *Drosophila*. In this model organism, *Yki* upregulation drives benign tissue overgrowth in imaginal discs. Our research demonstrates that the conserved metabolic regulator dPGC1 restricts Yki-driven tissue hyperplasia and helps maintain epithelial integrity *in vivo*. Combined *Yki* upregulation and *dPGC1* depletion results in tumors characterized by enlarged mitochondria and the upregulation of genes promoting mitochondrial fusion, a condition that is both necessary and sufficient for Yki-driven oncogenic growth. We further demonstrate that mitochondrial enlargement is associated with increased levels of the cell cycle regulator Cyclin E, which plays a critical role in tumor development. These findings identify dPGC1 as a context-dependent tumor suppressor that coordinates mitochondrial dynamics and cell cycle regulation in response to oncogene activation, with implications for understanding cancer development in humans.

## Introduction

Cancer is a multistep process involving the accumulation of mutations that affect different cellular functions known as the hallmarks of cancer. These include central processes such as cell proliferation, apoptosis, and cellular energetics. Together, these hallmarks promote the transformation of normal cells into malignant entities [1,2]. Tumors frequently undergo major changes in their metabolism to meet the high

**Data availability statement:** All relevant data are within the paper and its Supporting Information files.

**Funding:** This work was supported by the following grants awarded to H.H.: Danmarks Frie Forskningsfond (Grant No. 0134-00045B), https://dff.dk/. Novo Nordisk Fonden (Grant No. NNF18OC0052223), https://novonordiskfond-en.dk/. Neye Foundation, https://www.neye.dk/blog/neye-fonden/. Grosserer Alfred Nielsen og Hustru's Fond, https://grosserernielsenfonden.dk/. The funders had no role in study design, data collection and analysis, decision to publish, or preparation of the manuscript.

**Competing interests:** The authors have declared that no competing interests exist.

**Abbreviations :** CDKs, cyclin-dependent kinases; EM, electron microscopy; JNK, c-Jun N-terminal kinase; MMP, mitochondrial membrane potential; Mmp1, matrix metalloproteinase 1; PH3, phospho-Histone H3; PGC1, Peroxisome Proliferator Activated Receptor Gamma Coactivator-1; PRC, PGC1-related coactivator; TMRE, tetramethylrhodamine ethyl ester; YAP, Yes-associated protein; Yki, Yorkie.

energetic demands of cancer cells. Mitochondria are cytoplasmic organelles that play central roles in cellular energetics. Alterations in mitochondrial function can provide plasticity and adaptability to tumor cells growing in harsh environments, such as hypoxia, reduced nutrient availability, and exposure to cancer treatments [3]. Consequently, mitochondrial alterations influence various aspects of tumor initiation and progression, and genetic changes affecting mitochondria can increase cancer risk [4,5].

Mitochondria are highly dynamic organelles. Continuous fusion and fission events are crucial for maintaining mitochondrial integrity and function. Mitochondrial fission mediates the division of one mitochondrion into two and is essential for processes such as mitosis and quality control. Mitochondrial fusion, by contrast, merges two mitochondria into one, allowing the exchange of mitochondrial contents and supporting function, especially under cellular stress [6,7]. Beyond maintaining mitochondrial integrity and function, the balance between fission and fusion is crucial for regulating the cell cycle [8,9]. The cell cycle ensures that a cell replicates and segregates its genome faithfully before it divides into two identical daughter cells. Errors in this process can lead to genomic instability, which is a cancer driver [10]. Defective mitochondrial dynamics can result in an imbalance between fusion and fission, which has been associated with human diseases, including cancer. Therefore, mitochondrial dynamics must be tightly regulated to ensure proper cellular function and tissue homeostasis [11–16].

The Peroxisome Proliferator Activated Receptor Gamma Coactivator-1 (PGC1) family of transcriptional regulators plays central roles in controlling mitochondrial biogenesis and function. This family was originally characterized by the Spiegelman lab, who identified PGC1α as a cold-inducible coactivator of nuclear receptors involved in adaptive thermogenesis and mitochondrial regulation [17]. Subsequent work from the same group demonstrated that PGC1α orchestrates mitochondrial gene expression and respiration [18], regulates oxidative metabolism and muscle fiber type specification [19], and promotes mitochondrial biogenesis and oxidative phosphorylation in skeletal muscle cells [20]. In mammals, the PGC1 family includes PGC1α, PGC1β, and PGC1-related coactivator (PRC) [21,22]. These family members share structural and functional similarities. Among them, PGC1α has been most extensively studied in the context of cancer, where it regulates mitochondrial dynamics, metabolic reprogramming, and tumor progression. In contrast, PGC1β and PRC have primarily been characterized in normal physiology, where they regulate mitochondrial biogenesis. While PGC1β and PRC are primarily associated with cellular metabolism and energy homeostasis, PGC1β also contributes to mitochondrial fusion, notably through the regulation of Mfn2 and other mitochondrial factors [23].

PGC1α controls mitochondrial dynamics by regulating the expression of genes involved in fusion and fission, ensuring a proper balance between these processes to preserve mitochondrial integrity [24,25]. The role of PGC1α in cancer is complex and context-dependent. It can act as both a tumor suppressor and an oncogenic factor, depending on the cancer type and cellular context [26–31]. Therefore, elucidating the context-specific functions of PGC1α in cancer is crucial for developing targeted

therapies. In cancers where PGC1α acts as a tumor suppressor, strategies to enhance its activity might be beneficial. In contrast, in cancers where PGC1α supports tumor growth, inhibiting its function could be a potential therapeutic approach.

*Drosophila* is emerging as a valuable *in vivo* model to study various aspects of cancer [32]. Notably, *Drosophila* possesses a sole *PGC1* orthologue: *dPGC1*, also known as *spargel*, which has been involved in the regulation of energy metabolism and mitochondrial function [33,34]. The functional homology between dPGC1 and mammalian PGC1 members facilitates the study of PGC1 functions in *Drosophila* without the complications of redundancy.

The Hippo signaling pathway is a conserved regulator of normal and oncogenic growth. Its core components form a regulatory kinase cascade that, when active, limits tissue growth by phosphorylating and inhibiting the activity of transcriptional coactivator with PDZ-binding motif (TAZ) and Yes-associated protein (YAP), known as Yorkie (Yki) in *Drosophila* [35]. Deregulation of the activity of the Hippo pathway affects tumor development and has been associated with traits of oncogenesis such as cell proliferation and survival, cancer metabolism, invasion, and metastasis [36,37]. Consistent with their growth-regulatory roles, YAP and TAZ are frequently amplified, while upstream elements are mutated in different cancer types [38].

Work from Banerjee's group [39] demonstrated that activation of the Hippo pathway effector Yki in *Drosophila* leads to increased mitochondrial fusion through the transcriptional upregulation of key fusion genes such as *Mitofusin* (*dMfn,* also known as *Marf*) and *Optic atrophy 1* (*Opa1*). Their study showed that *Yki* overexpression alone is sufficient to induce mitochondrial elongation and to upregulate several mitochondrial genes, including those involved in fusion and oxidative stress responses, while notably not affecting *dPGC1* expression. These findings suggest that Yki can directly modulate mitochondrial structure and function independently of dPGC1, and that the regulation of mitochondrial fusion genes contributes to the growth-promoting role of Yki in epithelial tissues. Building on this foundation, our study explores how *dPGC1* depletion further modulates the transcriptional and morphological mitochondrial responses in a *Yki*-overexpressing context. By comparing *Yki* overexpression alone to *Yki* upregulation combined with *dPGC1* knockdown, we dissect the specific contributions of dPGC1 not only to mitochondrial dynamics and gene regulation, but also to the tumor-like growth properties driven by Yki activation.

We find that while depletion of the transcriptional coactivator *dPGC1* has a minor impact on normal *Drosophila* wing imaginal disc development, only causing a subtle growth defect, it dramatically drives tumor growth and cellular transformation in imaginal discs overexpressing *Yki*. This establishes dPGC1 as a context-dependent tumor suppressor that specifically limits Yki-driven tissue overgrowth without altering *Yki* expression or activity, and notably, not affecting overgrowth by other oncogenes. Tumors driven by *Yki* upregulation and *dPGC1* depletion exhibit enlarged mitochondria, a phenotype that correlates with the transcriptional upregulation of key mitochondrial fusion genes, *dMfn* and *Opa1*. We demonstrate that this is both necessary and sufficient to promote Yki-driven oncogenic growth and malignancy. Tumors driven by *Yki* overexpression and *dPGC1* knockdown show increased levels of Cyclin E protein. Cyclin E is essential for tumor development in this context, promoting growth and inducing DNA damage, two hallmarks of cancer. This study elucidates a novel connection between aberrant mitochondrial dynamics, defective cell cycle regulation, and DNA damage in oncogenic processes occurring *in vivo*, suggesting that targeting mitochondrial fusion components may offer new therapeutic strategies to limit Yki/YAP-driven tumor growth.

## Results

### dPGC1 is required for normal wing growth

The systemic functions of dPGC1 have been extensively studied, and it has been shown to control central physiological processes such as early development, tissue homeostasis, obesity, and aging [34,40–45]. However, its role in highly proliferative tissues such as developing organs or neoplastic growth remains poorly understood. To address this gap, we employed the *Drosophila* wing imaginal disc, a well-established *in vivo* model composed of a proliferative epithelial monolayer that gives rise to the adult wing and thorax during metamorphosis. This tissue serves as a powerful system for investigating epithelial development, growth regulation, and tumorigenesis [46].

PLOS Biology

To dissect the function of dPGC1 in this context, we examined the wings of *dPGC1¹* mutant flies, which carry a P-element insertion in the *dPGC1* gene. This insertion disrupts normal transcription and results in reduced gene function [47]. Despite this molecular alteration, *dPGC1¹* homozygous mutants are viable and do not display overt morphological abnormalities. These observations are consistent with studies in mice showing that PGC1α and PGC1β are not essential for normal development [20,48–51]. We detected a modest but statistically significant reduction in the size of both wing discs and adult wings in *dPGC1¹/dPGC1¹* mutants compared to controls (Figs 1A, 1B, and S1). These results indicate that dPGC1, while not essential for viability or gross morphology, plays a role in promoting normal wing growth. To confirm this role, we employed the Gal4-UAS binary system to manipulate *dPGC1* expression in a spatially and temporally controlled manner [52]. Using the MS1096-Gal4 driver, which is active in the wing disc epithelium (Fig 1C), we specifically modulated gene activity during wing development and assessed the resulting adult phenotypes. Consistent with the analysis of *dPGC1* mutants, expression of *dPGC1-RNAi* transgenes under the control of MS1096-Gal4 led to a modest size reduction (Fig 1D and 1E). The efficiency of the transgenes used to deplete *dPGC1* is shown in S2A Fig. Overexpression of *dPGC1* did not affect wing size or pattern (Fig 1D and 1E). These results indicate that, although dispensable for *Drosophila* viability, dPGC1 is required for normal wing growth.

## dPGC1 restricts Yki-driven tissue overgrowth in *Drosophila*

Given the role of dPGC1 in supporting normal wing growth, we next investigated whether this transcriptional regulator also influences pathological growth conditions. Since many of the molecular mechanisms that govern normal development are co-opted during tumorigenesis, we hypothesized that dPGC1 might modulate the cellular response to oncogenic signals. Among the various oncogenic drivers in *Drosophila*, we focused on the Hippo pathway effector Yki, a conserved

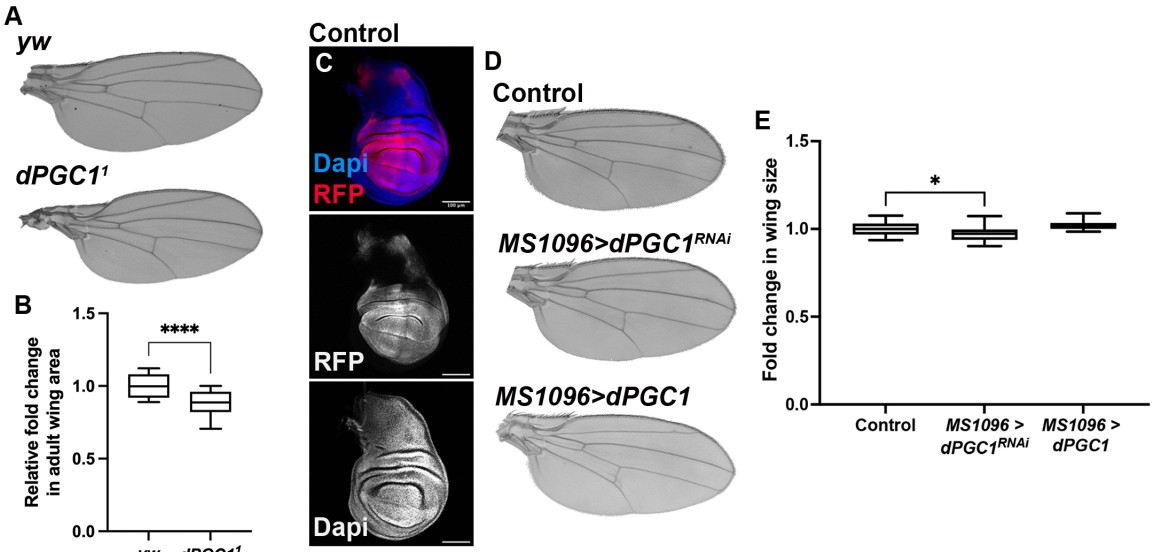

**Fig 1. Reduced dPGC1 activity leads to smaller wings. (A)** Cuticle preparations of adult wings from the following genotypes: *yw/yw* (control) and *dPGC1¹/dPGC1¹* (mutant). **(B)** Quantification of adult wing size of the genotypes indicated in A. Wing area is normalized to the mean of the control (*yw*). Statistical significance was determined using an unpaired *t* test (*n* = 20 [*yw*], *n* = 20 [*dPGC1¹*]). **** *p* < 0.0001. **(C)** Confocal micrograph showing a *MS1096-Gal4, UAS-RFP* third instar wing imaginal disc. RFP labels the region of the disc expressing this Gal4 line and is shown in red. DAPI labels the DNA and is shown in blue. Scale bar, 100 μm. **(D)** Cuticle preparations of adult wings from *MS1096-Gal4, UAS-LacZ* control flies; *MS1096-Gal4, UAS-dPGC1-RNAi* flies; and *MS1096-Gal4, EP-dPGC1* flies. **(E)** Quantification of adult wing size of the genotypes indicated in D. Wing area is normalized to the mean of the control (*MS1096 > LacZ*). Statistical significance was determined using unpaired *t* tests with Welch's correction (*n* = 20 [*MS1096 > LacZ*], *n* = 20 [*MS1096 > dPGC1-RNAi*], *n* = 19 [*MS1096 > dPGC1*]). **p* < 0.05. The data underlying all the graphs in the figure can be found in S2 Data.

proto-oncogene that promotes tissue overgrowth and modulates mitochondrial structure and function, processes in which dPGC1 is also implicated [39,47,53–57]. This functional overlap in mitochondrial regulation provided us with a strong rationale to explore whether dPGC1 modulates Yki-driven tumor-like growth. To test this, we depleted *dPGC1* in wing imaginal discs overexpressing *Yki* using the apterous-Gal4 (ap-Gal4) driver, which is commonly used to dissect various aspects of tumor formation [53,58–61]. As previously reported, *Yki* overexpression leads to a marked increase in tissue size (Fig 2A, 2C, and 2E) [55]. While *dPGC1* knockdown alone did not affect wing disc size, its depletion in the context of *Yki* overexpression further increased the overgrowth phenotype (Fig 2A–2E). These findings indicate that, although dPGC1 contributes modestly to normal wing development, it plays a critical role in restraining Yki-driven tissue expansion.

To complement the overexpression studies and assess the role of dPGC1 in a more physiologically relevant context, we analyzed *Warts* (*Wts*) mutant clones. Wts (Lats in mammals) is a core kinase in the Hippo pathway that phosphorylates and inhibits Yki, preventing its nuclear localization. As previously reported, loss of *Wts* mimics *Yki* overexpression and results in tissue overgrowth [55]. Consistent with this, *Wts* mutant clones were larger than control clones (Fig 2F, 2G, and 2I). Notably, *Wts* mutant clones expressing *dPGC1-RNAi* exhibited even greater overgrowth (Fig 2G–2I), supporting the idea that dPGC1 acts to restrain tissue expansion in cells with elevated Yki activity. These results support our previous observations and further reinforce a role for dPGC1 in limiting Yki-driven tissue overgrowth.

To determine whether the enhanced growth phenotype observed upon *dPGC1* depletion could be explained by increased *Yki* expression or activity, we used quantitative PCR (qPCR) to measure the transcript levels of *Yki* and its target genes *Cyclin E*, *Diap1*, and *bantam* in wing imaginal discs with or without *dPGC1* knockdown. These analyses revealed no significant changes, suggesting that the phenotype is not driven by elevated *Yki* expression or transcriptional activity (S3 Fig).

Given that dPGC1 modulates Yki-driven overgrowth, we asked whether it acts as a general regulator of oncogenic growth or plays a more specific role in restraining Yki-induced tumorigenesis. To address this, we downregulated *dPGC1* in tissues overexpressing the oncogenes *Epidermal Growth Factor Receptor* (*EGFR*) or the *Insulin Receptor* (*InR*). In both cases, *dPGC1* depletion did not enhance tissue overgrowth, in contrast to the effect observed in *Yki*-overexpressing discs (compare S4 Fig with Fig 2C–2E). This indicates that the tumor-suppressive role of dPGC1 is specifically relevant in the context of Yki-driven tumorigenesis rather than being a general response to oncogenic signaling.

## dPGC1 restrains oncogenic traits in Yki-induced tumorous discs

Having established that dPGC1 limits Yki-driven tissue overgrowth, we next asked whether it also suppresses other oncogenic traits associated with tumorigenesis. In addition to increased tissue size, malignant fly tumors often express the secreted matrix metalloproteinase 1 (Mmp1), which promotes cell migration and invasion by degrading the basement membrane [62,63]. While we did not detect Mmp1 expression in control discs, Mmp1 was observed in discrete patches of cells in discs overexpressing *Yki* (Fig 2J, 2K, and 2M). Notably, discs co-expressing *Yki* and *dPGC1-RNAi* displayed a robust increase in Mmp1 (Fig 2K–2M). *Mmp1* is a target gene of the c-Jun N-terminal kinase (JNK) pathway, a signaling cascade activated by internal and external stressors that regulates proliferation, apoptosis, and cell migration [64,65]. JNK acts as an oncogenic partner of pro-tumorigenic factors such as Ras$^{V12}$ and Yki [63,66]. In these contexts, JNK induces the accumulation of F-actin, which is observed in cells undergoing malignant transformation [42,63,67–70]. Consistent with these findings, we observed elevated F-actin levels in *Yki+dPGC1-RNAi* tumorous discs, along with increased expression of the JNK target gene *Mmp1* (Fig 2J–2L and 2N). These results were confirmed using an independent UAS-driven *dPGC1-shRNA* line (S5 Fig), whose knockdown efficiency is shown in S2A Fig.

In addition to invasive and structural changes, uncontrolled proliferation is a hallmark of oncogenic growth [2]. To determine whether this feature was also present in *Yki+dPGC1-RNAi* tumorous discs, we examined mitotic activity using phospho-Histone H3 (PH3) staining, a well-established marker of cells undergoing mitosis and thus a proxy for cell proliferation. This analysis revealed a significant increase in mitotic cells, indicating elevated proliferative activity and

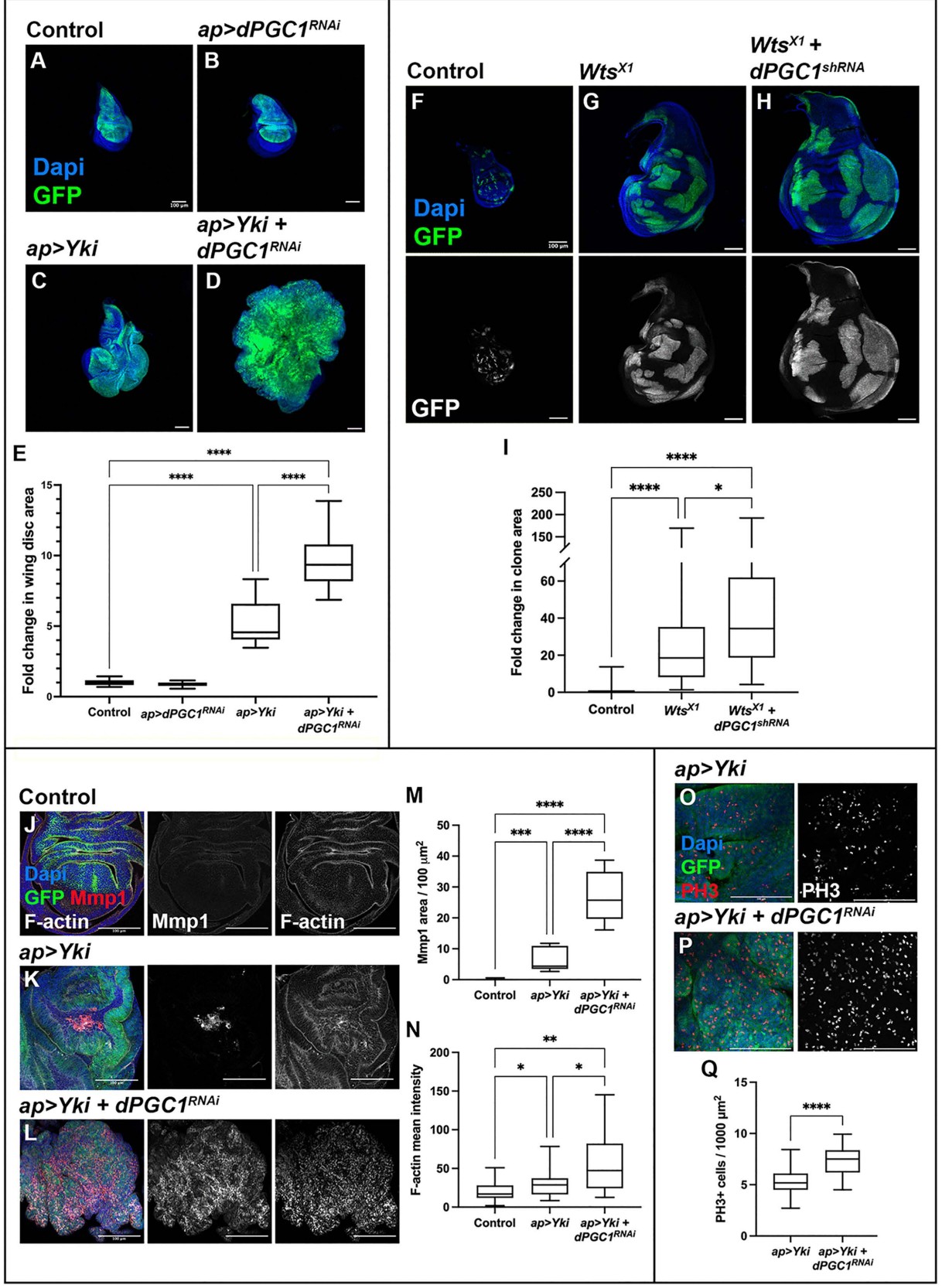

**Fig 2. Oncogenic growth in discs overexpressing *Yki* and downregulating *dPGC1*. (A–D)** Confocal micrographs showing discs of the following genotypes: *ap-Gal4, UAS-GFP, UAS-LacZ* (A); *ap-Gal4, UAS-GFP, UAS-dPGC1-RNAi* (B); *ap-Gal4, UAS-GFP, UAS-Yki, UAS-LacZ* (C); and *ap-Gal4, UAS-GFP, UAS-Yki, UAS-dPGC1-RNAi* (D). GFP is shown in green, and DAPI labels the DNA and is shown in blue. Scale bars, 100 μm. **(E)** Quantification of GFP-positive area in third instar wing imaginal discs of the genotypes shown in A–D. GFP-positive area is normalized to the mean of the control *(ap > LacZ)*. Statistical significance was determined using unpaired *t* tests with Welch's correction ($n = 18$ [*ap > LacZ*], $n = 19$ [*ap > dPGC1-RNAi*], $n = 20$ [*ap > Yki*], $n = 20$ [*ap > Yki, dPGC1-RNAi*]). ****$p < 0.0001$. **(F–H)** Confocal micrographs showing representative images of imaginal discs with GFP-labeled MARCM control clones (F), *Wts$^{X1}$* clones (G), and *Wts$^{X1}$* clones expressing *UAS-dPGC1-shRNA* (H). GFP is shown in green, and DAPI labels the DNA and is shown in blue. Scale bars, 100 μm. **(I)** Quantification of GFP-positive clones in the genotypes shown in F–H. GFP-positive area is normalized to the mean of the control. Statistical significance was determined using a Kruskal–Wallis test for non-parametric data ($n = 102$ [Control], $n = 92$ [*Wts$^{X1}$*], $n = 65$ [*Wts$^{X1}$ + dPGC1$^{shRNA}$*]). *$p < 0.05$, ****$p < 0.0001$. **(J–L)** Confocal micrographs of imaginal wing discs of the following genotypes: *ap-Gal4, UAS-GFP, UAS-LacZ* (J); *ap-Gal4, UAS-GFP, UAS-Yki, UAS-LacZ* (K); and *ap-Gal4, UAS-GFP, UAS-Yki, UAS-dPGC1-RNAi* (L). F-actin labels cell polarity and is shown in grayscale. Mmp1 is shown in red. GFP is shown in green. DAPI labels the DNA and is shown in blue. Scale bars, 100 μm. **(M)** Quantification of Mmp1 area per GFP-positive area in the genotypes shown in J–L. Statistical significance was determined using a Brown–Forsythe and Welch ANOVA test ($n = 12$ [*ap > LacZ*], $n = 11$ [*ap > Yki*], $n = 11$ [*ap > Yki, dPGC1-RNAi*]). ***$p < 0.001$, ****$p < 0.0001$. **(N)** Quantification of F-actin mean intensity in the genotypes shown in J–L. Statistical significance was determined using a Brown–Forsythe and Welch ANOVA test ($n = 31$ [*ap > LacZ*], $n = 22$ [*ap > Yki*], $n = 19$ [*ap > Yki, dPGC1-RNAi*]). *$p < 0.05$, **$p < 0.01$. **(O, P)** Confocal micrographs showing magnifications from tumorous wing discs with the following genotypes: *ap-Gal4, UAS-Yki, UAS-GFP, UAS-LacZ* (O) and *ap-Gal4, UAS-Yki, UAS-GFP, UAS-dPGC1-RNAi* (P). PH3 labels mitotic cells and is shown in red. GFP is shown in green. DAPI labels the DNA and is shown in blue. Scale bars, 10 μm. **(Q)** Quantification of number of PH3-positive cells per GFP-positive area in the genotypes shown in O and P. Statistical significance was determined using an unpaired *t* test ($n = 23$ [*ap > Yki*], $n = 19$ [*ap > Yki, dPGC1-RNAi*]). ****$p < 0.0001$. The data underlying all the graphs in the figure can be found in S2 Data.

highlighting an additional oncogenic feature of these tumors (Fig 2O–2Q). These findings suggest that *dPGC1* depletion not only enhances tissue overgrowth and promotes markers of malignancy but also accelerates cell cycle progression in the context of Yki activation.

In sum, our results show that *dPGC1* knockdown enhances Yki-induced overgrowth, leading to the formation of tumorous discs that exhibit markers of malignancy. Although *dPGC1* depletion does not have a major impact on normal growth, it is required to maintain controlled growth and tissue integrity in the context of *Yki* upregulation.

## Mitochondrial membrane potential in Yki-driven tumors

Given the central role of dPGC1 in maintaining mitochondrial activity, we evaluated mitochondrial functionality in tumors driven by Yki activation and *dPGC1* depletion. As a surrogate for mitochondrial function, we assessed mitochondrial membrane potential (MMP). MMP is a key indicator of mitochondrial function, reflecting the integrity of the electron transport chain and the capacity for ATP production via oxidative phosphorylation. To evaluate MMP, we employed tetramethylrhodamine ethyl ester (TMRE), a cell-permeant fluorescent dye that accumulates in mitochondria in proportion to their electrochemical gradient [71]. TMRE staining showed no significant difference in MMP between discs expressing *Yki* alone and those co-expressing *Yki* with *dPGC1-RNAi* (S6 Fig). These results indicate that *dPGC1* depletion does not impair MMP in Yki-driven tumors. This preservation of MMP may reflect compensatory metabolic adjustments that help maintain mitochondrial function despite structural remodeling, allowing tumor cells to sustain their energetic demands.

## dPGC1 loss selectively upregulates mitochondrial fusion genes in Yki-driven tumors

Given that MMP was not significantly affected by *dPGC1* depletion in Yki-driven tumors, we investigated whether this interaction altered the transcriptional landscape of genes involved in mitochondrial dynamics. This analysis was motivated by previous findings showing that PGC1 family proteins regulate the expression of genes involved in mitochondrial biogenesis and function [22] and that Yki modulates mitochondrial architecture to support tissue growth [39]. We examined whether *dPGC1* downregulation in the context of Yki activation affected the expression of genes involved in these processes. By using qPCR, we compared the mRNA levels of genes involved in different aspects of mitochondrial dynamics such as mitochondrial biogenesis, mitochondrial fusion and fission, mitophagy, and mitochondrial transport [11].

Among genes controlling mitochondrial biogenesis, we quantified the expression of *estrogen-related receptor* (*dERR*), *Ets at 97D* (*Ets97D*), and *erect wing* (*ewg*) [72–76]. We also assessed the expression of the genes *dMfn* and *Opa1*, which control fusion between outer and inner membranes, respectively; and *Dynamic-related protein 1* (*Drp1*), which mediates mitochondrial fission [77]. Additionally, we quantified the expression of genes involved in mitophagy, including *PTEN-induced putative kinase 1* (*Pink1*), *parkin* (*park*), and *Autophagy-related 7* (*Atg7*) [78,79]. Finally, we measured the levels of expression of genes involved in mitochondrial transport, such as *milton* (*milt*) and *mitochondrial Rho* (*miro*) [80,81]. We compared mRNA levels between *Yki*-upregulating discs (control) and *Yki + dPGC1-RNAi* tumorous discs (experimental condition). We found that discs co-expressing *Yki* and *dPGC1-RNAi* showed a significant increase in the expression of *dMfn* and *Opa1*, GTPases promoting mitochondrial fusion (Fig 3A). Interestingly, *milt* and *miro* were also upregulated in *Yki + dPGC1-RNAi* tumorous discs (Fig 3A). Although miro and milt play central roles in mitochondrial axonal transport, previous reports have suggested that they can also affect mitochondrial morphology [82–85]. For instance, *miro* overexpression in *Drosophila* models of Alzheimer's disease has been reported to cause an increased mitochondrial average length [84].

*dPGC1* depletion in otherwise normal discs did not affect the expression of genes controlling fusion/fission mechanisms to the same extent as in the context of *Yki* upregulation (S7 Fig). This observation suggests that the transcriptional response of mitochondrial dynamics genes to *dPGC1* depletion is influenced by Yki activity, indicating that mitochondrial remodeling becomes particularly responsive to *dPGC1* loss in an oncogenic environment driven by *Yki* upregulation.

To further explore the relationship between dPGC1 and mitochondrial fusion in the context of *Yki* upregulation, we analyzed the expression of *dMfn* and *Opa1* in discs overexpressing *dPGC1* alongside *Yki*. In contrast to the effects observed upon *dPGC1* downregulation, *dPGC1* upregulation in *Yki*-overexpressing discs did not significantly affect tissue growth and did not alter *dMfn* and *Opa1* mRNA levels (S8 Fig). These findings suggest that the transcriptional upregulation of mitochondrial fusion genes is a specific consequence of *dPGC1* knockdown in a Yki-activated context, highlighting a sensitized mitochondrial response to Yki signaling rather than a general effect of altered dPGC1 levels.

## Functional analysis of mitochondrial fusion genes in wing disc development

The genes *dMfn, Opa1, milt,* and *miro*, which were upregulated in *Yki + dPGC1-RNAi* tumorous discs, have been extensively studied in various model systems [11,86–88]. However, their specific functions in regulating mitochondrial morphology in the *Drosophila* wing imaginal disc remain uncharacterized. To address this, we experimentally manipulated the expression of these genes during wing disc development. To visualize mitochondria, we used a mitochondrial-targeted GFP (Mito-GFP) transgene, which contains localization sequences that direct GFP to mitochondria [89].

The wing imaginal disc is composed of two distinct epithelial layers: a densely packed columnar epithelium referred to as the main epithelium, and a squamous epithelium known as the peripodial membrane [90]. The compact architecture and limited cytoplasmic volume of the main epithelium posed challenges for high-resolution imaging of subcellular structures such as mitochondria (Fig 3B). To circumvent these limitations, we performed mitochondrial imaging in the peripodial membrane, which consists of large, flattened cells with expanded cytoplasmic space that facilitates clear visualization of mitochondrial morphology. In this context, mitochondria appeared as distinct and well-defined organelles (Fig 3C), enabling a reliable assessment of morphological changes following genetic manipulations.

To assess the roles of genes upregulated in *Yki + dPGC1-RNAi* tumors, we used the Grunge-Gal4 (Gug-Gal4) driver to manipulate *dMfn, Opa1*, and *miro* specifically in the peripodial membrane of the wing imaginal disc [91]. This allowed us to visualize mitochondrial changes resulting from their experimental deregulation. dMfn and Opa1 are central regulators of mitochondrial fusion. Upregulation of *dMfn* led to a connected mitochondrial network, indicative of enhanced fusion (Fig 3D; quantified in S9 Fig). In contrast, overexpression of *Opa1* produced a subtler phenotype and mitochondria appeared slightly elongated compared to controls, but the overall impact on network connectivity was modest relative to *dMfn* overexpression (compare Fig 3F with Fig 3D; quantified in S9 Fig). Depletion of *dMfn1* or *Opa1* caused mitochondrial

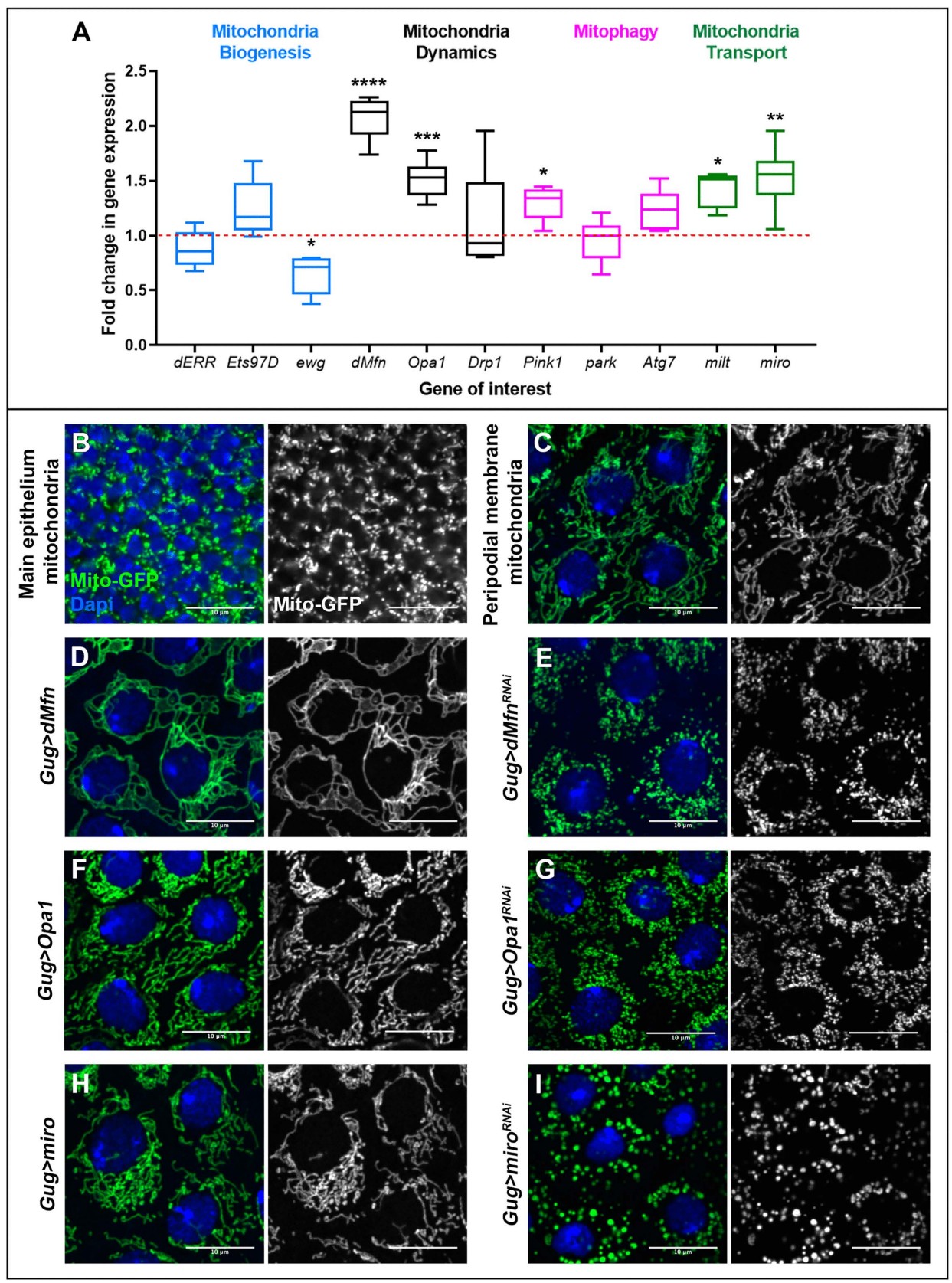

**Fig 3. Aberrant mitochondrial dynamics in *Yki + dPGC1-RNAi* tumorous discs. (A)** mRNA quantification by qPCR of the indicated genes in *ap-Gal4, UAS-GFP, UAS-Yki, UAS-dPGC1-RNAi* wing imaginal discs using *ap-Gal4, UAS-GFP, UAS-Yki, UAS-LacZ* as the control genotype. Genes are separated in different categories: mitochondria biogenesis (blue), mitochondria dynamics (black), mitophagy (pink), and mitochondria transport (green). Each genotype was run in five to six biological replicates. RP49 was used as the housekeeping gene. Statistical significance was determined using unpaired $t$ tests with Welch's correction for parametric data and Mann-Whitney tests for non-parametric data. *$p < 0.05$, **$p < 0.01$, ***$p < 0.001$, ****$p < 0.0001$. The data underlying this graph can be found in S2 Data. **(B)** Confocal micrograph of the main epithelium of an *ap-Gal4, UAS-Mito-GFP* wing imaginal disc. Mitochondria are labeled with Mito-GFP and are shown in green. DAPI is shown in blue. Scale bar, 10 µm. **(C)** Confocal micrograph of the peripodial membrane of a *Gug-Gal4, UAS-Mito-GFP* wing imaginal disc. Mitochondria are labeled with Mito-GFP and are shown in green. DAPI is shown in blue. Scale bar, 10 µm. **(D–I)** Confocal micrographs of the peripodial membrane in the following genotypes: *Gug-Gal4, UAS-Mito-GFP, UAS-dMfn* (D); *Gug-Gal4, UAS-Mito-GFP, UAS-dMfn-RNAi* (E); *Gug-Gal4, UAS-Mito-GFP, EP-Opa1* (F); *Gug-Gal4, UAS-Mito-GFP, UAS-Opa1-RNAi* (G); *Gug-Gal4, UAS-Mito-GFP, UAS-miro* (H); and *Gug-Gal4, UAS-Mito-GFP, UAS-miro-RNAi* (I). Mitochondria are labeled with Mito-GFP and are shown in green. DAPI is shown in blue. Scale bars, 10 µm.

fragmentation (Fig 3E and 3G; quantified in S9 Fig). The phenotypic differences observed when *dMfn* and *Opa1* were overexpressed likely reflect the distinct molecular roles of dMfn and Opa1 within the mitochondrial fusion machinery. While both proteins promote fusion, dMfn primarily facilitates the merging of the outer mitochondrial membrane, a process that directly contributes to the formation of interconnected networks. Opa1 regulates inner membrane fusion and cristae architecture, which may lead to elongation without extensive network formation.

*milt* and *miro* were also upregulated in *Yki + dPGC1-RNAi* tumors. Although milt and miro are traditionally associated with mitochondrial transport, recent studies suggest their involvement in mitochondrial dynamics [82–85]. In our expression analysis, *miro* showed the highest level of upregulation between the two (Fig 3A), making it a particularly compelling candidate for functional investigation. Altering *miro* expression resulted in noticeable changes in mitochondrial morphology (Fig 3H and 3I; quantified in S9 Fig). *miro* overexpression resulted in mitochondria that appeared elongated but not as extensively fused as those observed in *dMfn*-overexpressing imaginal discs (Fig 3H; quantified in S9 Fig). The most pronounced effects were observed upon *miro* depletion. In these cells, mitochondria appeared smaller and more condensed, as indicated by the increased intensity and compaction of the Mito-GFP signal (Fig 3I; quantified in S9 Fig). This phenotype differs qualitatively from the fragmentation seen with *dMfn* or *Opa1* knockdown and may reflect a different role in mitochondrial morphology. Given its established function in mitochondrial transport, *miro* downregulation may disrupt mitochondrial positioning and cytoskeletal interactions, leading to clustering rather than classical fragmentation.

The efficiency of the transgenes used to upregulate and knock down *dMfn, Opa1*, and *miro* is shown in S2B–S2D Fig.

## Enhanced mitochondrial fusion in Yki-driven tumors upon *dPGC1* depletion

Our analyses revealed transcriptional upregulation of mitochondrial fusion genes in *Yki + dPGC1-RNAi* tumors, suggesting a shift in mitochondrial dynamics. To investigate whether these transcriptional changes led to altered mitochondrial morphology, we examined mitochondria in the tumorous tissue. Since the ap-Gal4 driver used to induce tumor formation was active in the main epithelium, which contains small, densely packed cells with limited cytoplasmic space, confocal imaging of mitochondria in this tissue was technically challenging (Fig 3B). To overcome this limitation and obtain high-resolution structural information, we employed electron microscopy (EM) to assess mitochondrial morphology in Yki-driven tumors with or without *dPGC1* depletion.

Consistent with the upregulation of mitochondrial fusion genes such as *dMfn* and *Opa1*, we observed that mitochondria in *Yki + dPGC1-RNAi* tumorous discs were larger and more elongated than those in *Yki*-expressing discs (Fig 4A, 4B, 4D, and 4E). Next, we assessed the consequences of downregulating *dMfn*, the mitochondrial fusion gene showing the highest upregulation in *Yki + dPGC1-RNAi* tumorous discs (Fig 3A). We found that mitochondria in tumors with reduced *dMfn* appeared larger and more rounded compared to those in *Yki + dPGC1-RNAi* tumors (Fig 4B–4E). Notably, these swollen mitochondria exhibited disrupted cristae, with some displaying a characteristic onion-like ultrastructure (Fig 4C and 4C′).

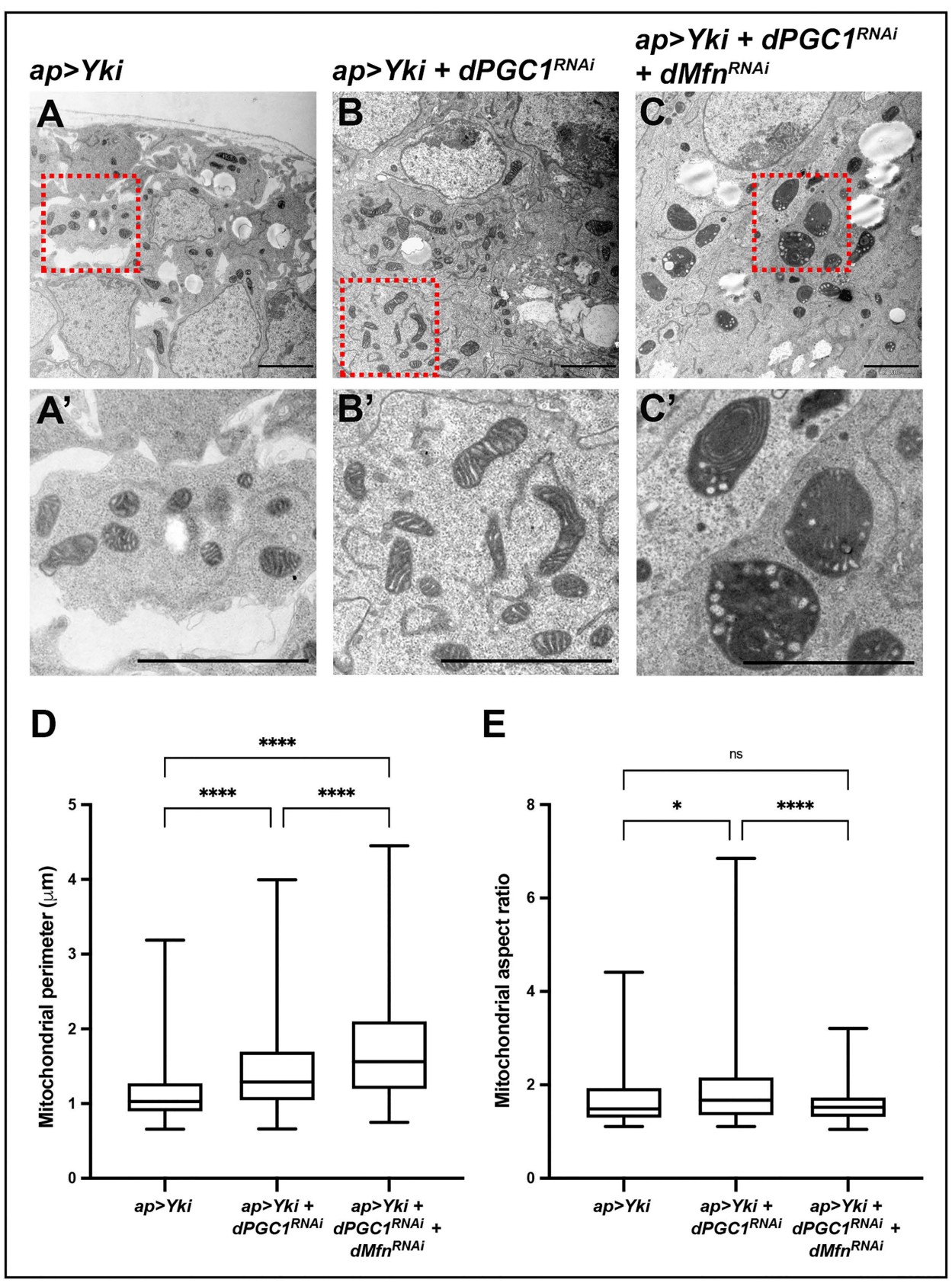

**Fig 4. dPGC1 and dMfn regulate mitochondrial morphology in Yki-driven tumors. (A–C)** Representative electron microscopy micrographs obtained from discs of the following genotypes: *ap-Gal4, UAS-GFP, UAS-Yki, UAS-LacZ* (A); *ap-Gal4, UAS-GFP, UAS-Yki, UAS-dPGC1-RNAi* (B); and *ap-Gal4, UAS-GFP, UAS-Yki, UAS-dPGC1-RNAi, UAS-dMfn-RNAi* (C). The dashed red squares in A–C indicate the regions that are shown as magnifications in A′–C′. Scale bars, 2 μm. **(D, E)** Quantification of mitochondrial perimeter (D) and mitochondrial aspect ratio (E) in the genotypes shown in A–C. Statistical significance was determined using one-way ANOVA tests ($n = 164$ [*ap > Yki*], $n = 214$ [*ap > Yki, dPGC1-RNAi*], $n = 218$ [*ap > Yki, dPGC1-RNAi, dMfn-RNAi*]). ns, non-significant. *$p < 0.05$, ****$p < 0.0001$. The data underlying all the graphs in the figure can be found in S2 Data.

In addition, we observed vesicle-like structures in those mitochondria, further suggesting that cristae organization and mitochondrial integrity were compromised upon *dMfn* depletion (Fig 4C and 4C′). These structural abnormalities suggest that cristae disorganization may impair mitochondrial function, potentially disrupting energy production and cellular homeostasis, which could in turn limit the ability of cells to sustain oncogenic growth.

## Contribution of mitochondrial fusion genes to Yki-mediated tumor growth

Previous studies showed that Yki upregulates genes involved in mitochondrial fusion, a process that plays a central role in its growth-promoting activity [39]. Building on this, we have demonstrated that depletion of *dPGC1* further enhances Yki-induced tissue overgrowth, with the resulting tumorous discs exhibiting elongated mitochondria. These findings led us to hypothesize that the upregulation of mitochondrial fusion genes may contribute to tumor formation when *Yki* is overexpressed and *dPGC1* is knocked down. If this is the case, then reducing the expression of these fusion-promoting genes should mitigate tumor growth. To test this, we used *UAS-RNAi* transgenes targeting *dMfn* or *Opa1* in *Yki + dPGC1*-depleted tumors, which significantly reduced tissue growth compared to control tumors (Fig 5A–5D and 5G). These results indicate that mitochondrial fusion supports tumor expansion in this context. Next, we investigated whether upregulation of mitochondrial fusion genes alone is sufficient to drive tumorigenesis in cooperation with the proto-oncogene Yki. Co-expression of *Yki* with either *dMfn* or *Opa1* led to a marked increase in disc size (Fig 5A and 5E–5G), indicating that enhanced mitochondrial fusion can potentiate Yki-driven tissue overgrowth. Importantly, these tumors exhibited hallmark features of malignancy, including elevated expression of *Mmp1* and disrupted epithelial architecture, as revealed by F-actin staining (Fig 5H and 5I). In contrast, knockdown or overexpression of these fusion genes in otherwise normal discs resulted in only minor changes in tissue size, effects that were significantly less pronounced than those observed in the tumor context (S10 Fig). Our findings highlight that dMfn and Opa1 are critical modulators of Yki-driven tumorigenesis.

To further explore the oncogenic contribution of the genes upregulated in our expression analysis (Fig 3A), we examined the function of miro. We found that, in contrast to *dMfn* and *Opa1*, *miro* upregulation was not sufficient to drive tumorigenesis in combination with Yki. Moreover, *miro* depletion did not alter the growth of *Yki + dPGC1-RNAi* tumors (S11 Fig).

In summary, our results demonstrate that genes controlling mitochondrial morphology play a gene-specific and context-dependent role in Yki-mediated tumorigenesis. While upregulation of fusion genes such as *dMfn* and *Opa1* synergizes with *Yki* to promote tissue overgrowth and malignancy, *miro* does not exhibit the same oncogenic potential.

## Upregulation of Cyclin E in tumors caused by *Yki* upregulation and *dPGC1* depletion

Mounting evidence has established a connection between mitochondrial dynamics and the cell cycle [92]. Proper control of the cell cycle is essential for accurate genome transmission during cell division, and its disruption can lead to genomic instability, which is a hallmark of cancer. [1,2]. Cyclin-dependent kinases (CDKs) and their regulatory cyclins are well-established drivers of cell cycle progression [93,94]. Beyond these canonical regulators, mitochondria have emerged as key modulators of cell cycle control, with their dynamic behavior contributing to the maintenance of genomic stability [95–98].

Studies in mammalian cells and in *Drosophila* showed that cells with highly fused mitochondria exhibit an elevation in the levels of the oncogene *Cyclin E* [9,99–102]. Cyclin E peaks at the G1-S transition, activating CDK2 and promoting S phase entry [103]. Aberrant Cyclin E levels can trigger premature entry into S phase, leading to replicative stress and DNA damage, which are key contributors to genomic instability in cancer [104–106].

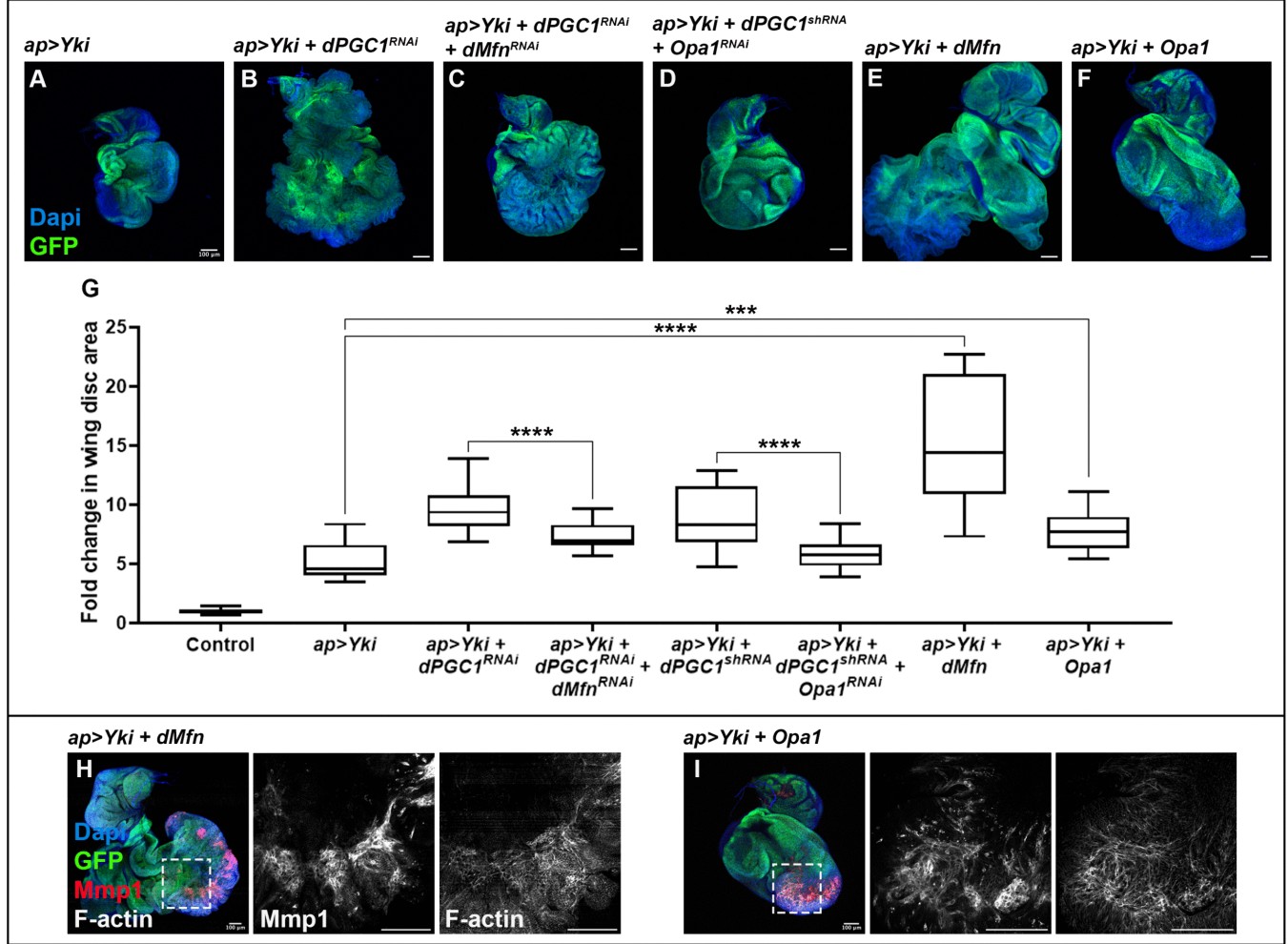

**Fig 5. dMfn and Opa1 synergize with Yki to promote tumor growth and malignancy. (A–F)** Confocal micrographs showing discs of the following genotypes: *ap-Gal4, UAS-GFP, UAS-Yki, UAS-LacZ* (A); *ap-Gal4, UAS-GFP, UAS-Yki, UAS-dPGC1-RNAi* (B); *ap-Gal4, UAS-GFP, UAS-Yki, UAS-dPGC1-RNAi, UAS-dMfn-RNAi* (C); *ap-Gal4, UAS-GFP, UAS-Yki, UAS-dPGC1-shRNA, UAS-Opa1-RNAi* (D); *ap-Gal4, UAS-GFP, UAS-Yki, UAS-dMfn* (E); and *ap-Gal4, UAS-GFP, UAS-Yki, EP-Opa1* (F). GFP is shown in green, and DAPI labels the DNA and is shown in blue. Scale bars, 100 μm. **(G)** Quantification of GFP-positive area in third instar wing imaginal discs of the genotypes shown in A–F. GFP-positive areas are normalized to the mean of the control (*ap>LacZ*). Statistical significance was determined using unpaired *t* tests with Welch's correction ($n = 18$ [*ap>LacZ*], $n = 20$ [*ap>Yki*], $n = 20$ [*ap>Yki, dPGC1-RNAi*], $n = 20$ [*ap>Yki, dPGC1-RNAi, dMfn-RNAi*], $n = 20$ [*ap>Yki, dPGC1-shRNA*], $n = 20$ [*ap>Yki, dPGC1-shRNA, Opa1-RNAi*], $n = 13$ [*ap>Yki, dMfn*], $n = 12$ [*ap>Yki, Opa1*]). ***$p < 0.001$, ****$p < 0.0001$. The data underlying this graph can be found in S2 Data. **(H, I)** Confocal micrographs of imaginal discs of the following genotypes: *ap-Gal4, UAS-GFP, UAS-Yki, UAS-dMfn* (H) and *ap-Gal4, UAS-GFP, UAS-Yki, UAS-Opa1* (I). F-actin labels cell polarity and epithelial organization and is shown in grayscale. Mmp1 is shown in red. GFP is shown in green. DAPI labels the DNA and is shown in blue. Scale bars, 100 μm.

We have previously shown that *dPGC1* depletion in epithelial cells with elevated Yki signaling led to an increase in mitochondrial size, which correlated with the transcriptional upregulation of genes promoting mitochondrial fusion, and is associated with tumor formation. These observations prompted us to investigate whether *Yki + dPGC1-RNAi* tumors exhibit abnormal levels of Cyclin E. We first examined *Cyclin E* mRNA levels and found no significant differences between discs expressing *Yki* alone and those co-expressing *Yki* with *dPGC1-RNAi* (Fig 6A). Interestingly, although transcript levels remained unchanged, Cyclin E protein levels were elevated in tumors driven by *Yki* overexpression and *dPGC1* depletion (Figs 6B, 6C, and S12). These results indicate that Cyclin E upregulation occurs post-transcriptionally and

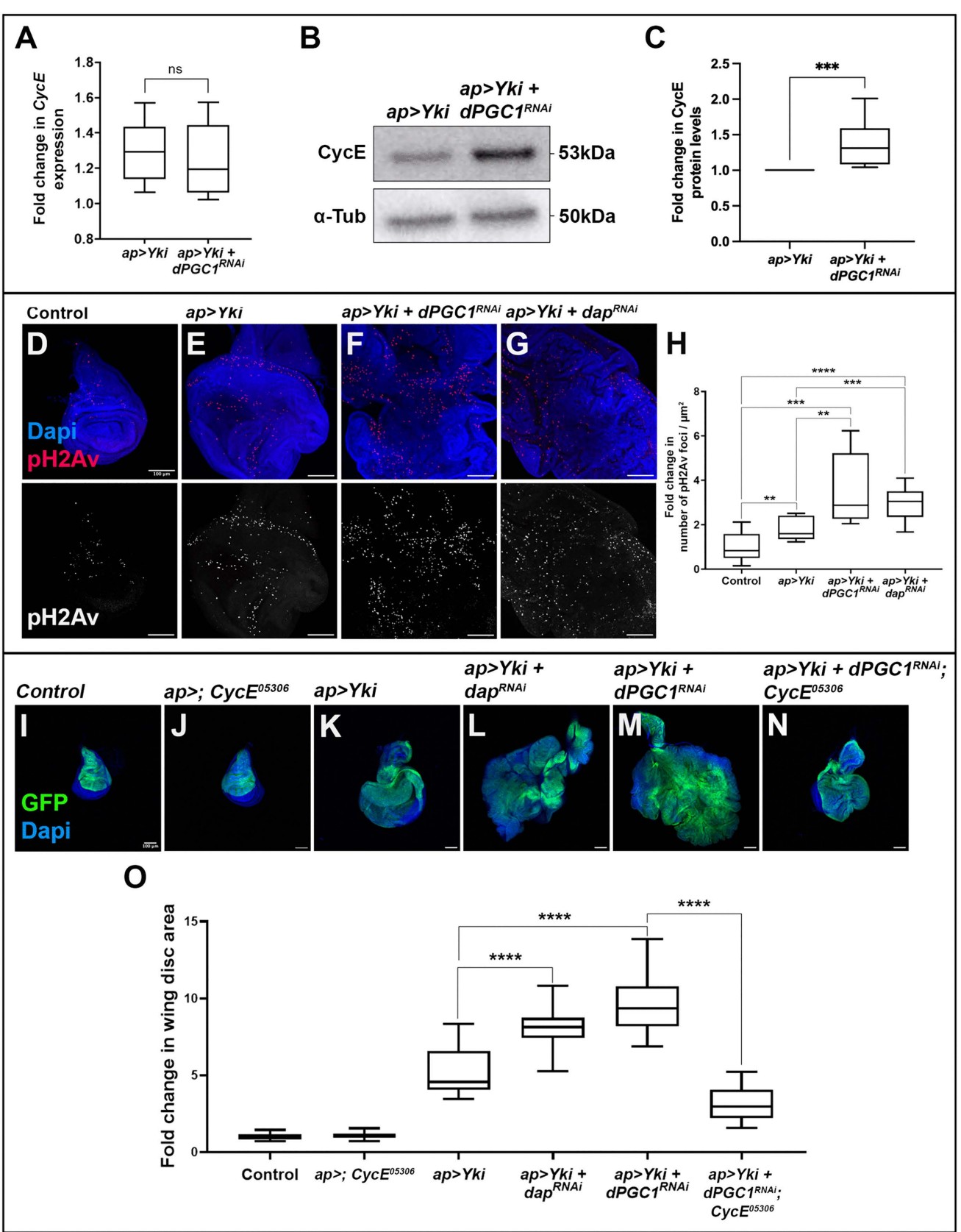

**Fig 6. Cyclin E stabilization drives tumor growth and DNA damage in *Yki + dPGC1-RNAi* discs. (A)** Quantification of *Cyclin E* mRNA by qPCR in imaginal discs of the following genotypes: *ap-Gal4, UAS-GFP, UAS-Yki, UAS-LacZ* (*ap > Yki*) and *ap-Gal4, UAS-GFP, UAS-Yki, UAS-dPGC1-RNAi* (*ap > Yki, dPGC1-RNAi*). Each genotype was run in four to five biological replicates. RP49 was used as the housekeeping gene. Statistical significance was determined using an unpaired *t* test with Welch's correction. ns, non-significant. **(B)** Western blot showing Cyclin E protein levels in third instar wing imaginal discs of the indicated genotypes. α-Tubulin was used as the loading control. **(C)** Relative quantification of Cyclin E protein levels in wing imaginal discs of the genotypes shown in B. Cyclin E protein levels were normalized to α-Tubulin protein levels for each western blot membrane. The obtained Cyclin E/α-Tubulin ratio was normalized to that of the control (*ap > Yki*). Statistical significance was determined using a Mann–Whitney test (n = 7). ***p < 0.001. **(D–G)** Maximum projection of confocal micrographs of imaginal discs of the following genotypes: *ap-Gal4, UAS-GFP, UAS-LacZ* (D); *ap-Gal4, UAS-GFP, UAS-Yki, UAS-LacZ* (E); *ap-Gal4, UAS-GFP, UAS-Yki, UAS-dPGC1-RNAi* (F); and *ap-Gal4, UAS-GFP, UAS-Yki, UAS-dap-RNAi* (G). pH2Av is shown in red and DAPI is shown in blue. Scale bars, 100 μm. **(H)** Quantification of pH2Av-positive foci in the genotypes shown in D-G. Data was normalized to the mean of the control (*ap > LacZ*). Statistical significance was determined using unpaired *t* tests with Welch's correction (n = 12 [*ap > LacZ*], n = 12 [*ap > Yki*], n = 12 [*ap > Yki, dPGC1-RNAi*], n = 20 [*ap > Yki, dap-RNAi*]). **p < 0.01, **p < 0.001, ****p < 0.0001. **(I–N)** Confocal micrographs showing discs of the following genotypes: *ap-Gal4, UAS-GFP, UAS-LacZ* (I); *ap-Gal4, UAS-GFP/Cyclin E-05306* (J); *ap-Gal4, UAS-GFP, UAS-Yki, UAS-LacZ* (K); *ap-Gal4, UAS-GFP, UAS-Yki, UAS-dap-RNAi* (L), *ap-Gal4, UAS-GFP, UAS-Yki, UAS-dPGC1-RNAi* (M); and *ap-Gal4, UAS-GFP, UAS-Yki, UAS-dPGC1-RNAi/Cyclin E-05306* (N). GFP is shown in green, and DAPI labels the DNA and is shown in blue. Scale bars, 100 μm. **(O)** Quantification of GFP-positive area in third instar wing imaginal discs of the genotypes shown in I–N. GFP-positive area is normalized to the mean of the control (*ap > LacZ*). Statistical significance was determined using unpaired *t* tests with Welch's correction (n = 18 [*ap > LacZ*], n = 20 [*ap >; CycE-05306*], n = 20 [*ap > Yki*], n = 20 [*ap > Yki, dap-RNAi*], n = 20 [*ap > Yki, dPGC1-RNAi*], n = 20 [*ap > Yki, dPGC1-RNAi; CycE-05306*]). ****p < 0.0001. The data underlying all the graphs in the figure can be found in S2 Data.

are consistent with previous studies linking mitochondria to Cyclin E regulation through non-transcriptional mechanisms [97,98].

Aberrant Cyclin E levels can lead to replicative stress and DNA damage in human cells and *Drosophila* [105,107,108]. In response to DNA damage, checkpoint proteins phosphorylate the histone H2AX (H2Av in *Drosophila*) to mark the sites of damage [109,110]. Antibodies recognizing the phosphorylated H2Av (pH2Av) can serve as markers to detect DNA damage. As expected, *Yki* overexpression led to an increase in pH2Av-positive foci (Fig 6D, 6E, and 6H). Notably, this effect was enhanced when *dPGC1* was knocked down (Fig 6E, 6F, and 6H), which is consistent with an increase in Cyclin E. These observations show that *dPGC1* depletion in discs activating the *Yki* proto-oncogene leads to an increase in Cyclin E levels and DNA damage.

Oncogene-induced DNA damage is a mutagenic source in cancer, and DNA damage can promote tumorigenesis in *Drosophila* [111–113]. We then analyzed whether Cyclin E upregulation contributes to the growth of *Yki + dPGC1-RNAi* tumors. To investigate the role of Cyclin E dosage in tumor growth, we induced the formation *Yki + dPGC1-RNAi* tumors in a *Cyclin E* heterozygous (+/−) background and analyzed whether reduced Cyclin E levels affected tumor development. We observed that *Cyclin E* heterozygous discs did not display any growth defects (Fig 6I, 6J, and 6O). However, when *Yki* was overexpressed and *dPGC1* was depleted in a *Cyclin E* heterozygous background, tumor size was dramatically reduced, indicating that reduced Cyclin E dosage specifically impairs tumor growth under these conditions (Fig 6M–6O). This suggests that Cyclin E upregulation is a central factor driving tumorigenesis in *Yki + dPGC1-RNAi* tumors.

Next, we asked whether upregulation of Cyclin E was sufficient to enhance Yki-driven growth. Cyclin E activity is negatively regulated by the Cyclin E-dependent kinase inhibitor p21/dacapo (dap), which represses Cyclin E/CDK2 function. Therefore, reduced levels of dap increase Cyclin E/CDK2 activity [114,115]. To test this, we depleted *dap* in *Yki*-expressing discs and found that this was sufficient to significantly enhance tumor growth (Fig 6K, 6L, and 6O), supporting the idea that elevated Cyclin E activity promotes Yki-driven tumorigenesis. Interestingly, similar to what we observed in the *Yki + dPGC1-RNAi* context, *Yki + dap-RNAi* tumors also exhibited increased levels of DNA damage (Fig 6E, 6G, and 6H), suggesting that Cyclin E upregulation may contribute to genomic instability in these tumors. The efficiency of *UAS-dap-RNAi* is shown in S13 Fig. In sum, these results reveal that *dPGC1* downregulation in the context of Yki activation leads to Cyclin E upregulation, which promotes the accumulation of DNA damage and drives increased tumor growth.

Finally, to determine whether the increase in Cyclin E is a general consequence of *dPGC1* depletion or specific to the oncogenic context of Yki activation, we analyzed Cyclin E levels in *dPGC1¹* homozygous mutant wing discs (−/−) in the

absence of *Yki* overexpression. We did not observe any detectable changes in Cyclin E protein levels under these conditions (S14 Fig). This indicates that *dPGC1* depletion alone is not sufficient to stabilize Cyclin E and further supports the notion that the oncogenic traits observed, such as Cyclin E accumulation, are specific to the context of *Yki* upregulation.

## Discussion

Our findings show that dPGC1 functions as a tumor suppressor by limiting Yki-driven tissue overgrowth in the *Drosophila* wing imaginal disc. Under normal conditions, *dPGC1* mutants exhibit only a minor growth defect, and altering dPGC1 levels during physiological wing development does not lead to overt phenotypes. This suggests that dPGC1 is largely dispensable for normal tissue growth but becomes critical under oncogenic stress, particularly when Yki is hyperactivated. These observations align with recent work by the Halder group, which showed that Yki/YAP transcriptional input is not required for normal growth but instead drives ectopic tissue expansion through the induction of an aberrant genetic program [116]. Such aberrant activation represents a pathological stress condition, further supporting the idea that dPGC1 is required to maintain tissue integrity under oncogenic challenges. Due to the lack of a specific antibody against dPGC1, our conclusions are based on transcript and genetic evidence, which limits direct assessment of protein-level regulation in this context.

This selective requirement becomes even more striking when considering its specificity. Although dPGC1 does not modulate tumor-like growth driven by other oncogenic signals such as EGFR or InR, it plays a crucial role in suppressing Yki-induced overgrowth. Importantly, this suppression occurs without altering Yki expression or activity, suggesting that dPGC1 acts downstream or in parallel to Yki signaling. In this context, dPGC1 influences the transcription of mitochondrial fusion genes, pointing to a role in modulating mitochondrial dynamics in response to Yki-driven oncogenic stress. This is consistent with previous findings that link changes in mitochondrial morphology to the growth-promoting effects of Yki [39], reinforcing a model in which dPGC1 safeguards tissue integrity by regulating mitochondrial function under pathological conditions.

This context-dependent role is consistent with observations in mammalian systems. For example, *PGC1α* knockout mice develop normally, indicating that PGC1 family members are not required for normal development [49]. However, under stress conditions, *PGC1α* mutant mice exhibit a range of physiological defects, highlighting the importance of PGC1α in maintaining cellular homeostasis during pathological challenges [20,117]. A similar context-specific function has been described for PGC1α in oncogenic processes. Its expression is frequently downregulated in various tumor types, suggesting a tumor-suppressive role [30,118–120]. Conversely, other studies have shown that PGC1α can promote tumorigenesis and metastasis in certain cellular and tissue contexts [26,121]. Together, these findings highlight the importance of the cellular context in determining the pro- or antitumorigenic roles of these transcriptional regulators.

Yki promotes tissue overgrowth by regulating gene expression through its interaction with the TEAD transcription factor Scalloped, inducing targets such as *Cyclin E* and mitochondrial fusion genes like *dMfn* and *Opa1* [39,122–124]. Our data show that *dPGC1* depletion in the context of Yki hyperactivation leads to upregulation of mitochondrial fusion genes, enlarged mitochondria, and enhanced tumor growth. These observations align with previous studies showing that increased mitochondrial fusion supports proliferation in Ras-transformed fibroblasts and *Drosophila* stem cells [125–127]. Importantly, this regulation appears to be context-dependent, as *dPGC1* depletion in otherwise normal discs has little or no detectable impact on the expression of mitochondrial fusion and fission genes. These findings suggest that dPGC1 dampens the ability of Yki to induce the expression of genes promoting mitochondrial fusion, thereby limiting oncogenic proliferation. Supporting this, reducing dMfn levels in *Yki + dPGC1-RNAi* discs disrupts mitochondrial morphology and impairs tumor growth. EM analyses of these discs reveal swollen mitochondria with disorganized cristae, structures essential for respiration and ATP production. Such defects likely compromise mitochondrial function and contribute to reduced tumor expansion. Beyond their roles in fusion, Opa1 and Mfn2 also regulate cristae architecture and mitochondrial-ER contacts, which are critical for apoptosis, oxidative phosphorylation, and

calcium signaling [128,129]. While our results support a role for enhanced mitochondrial fusion in Yki-driven tumorigenesis, we acknowledge that mitochondrial enlargement may also result from reduced fission activity or imbalances in mitochondrial biogenesis and mitophagy. Future studies employing live-cell imaging and dynamic assays will be essential to fully characterize the mitochondrial remodeling events underlying oncogenic transformation in this context. These broader functions may help explain how altered mitochondrial structure and inter-organelle communication contribute to the oncogenic phenotype.

Alterations in mitochondrial structure and function can influence key cell cycle regulators, linking metabolic and structural changes to proliferative outcomes [96–98,100,102]. Mitochondrial fusion has been shown to stabilize and increase Cyclin E levels, promoting G1–S progression [99,100]. Consistent with this, our results demonstrate that *dPGC1* depletion in the context of *Yki* upregulation leads to increased Cyclin E protein levels. This correlates with elevated DNA damage, a hallmark of genomic instability commonly observed in tumors [10]. We provide experimental evidence that Cyclin E upregulation is essential for tumor growth in this setting, identifying it as a central oncogenic driver in *Yki + dPGC1* knockdown tumors. We do not detect changes in *Cyclin E* mRNA levels, and the observed Cyclin E accumulation likely reflects post-transcriptional stabilization mechanisms. Mitochondrial dysfunction can activate retrograde signaling via ROS, leading to modulation of cell cycle regulators such as p53 and Dap, which indirectly affect Cyclin E turnover [97,98]. These mechanisms provide a plausible link between mitochondrial state and Cyclin E stabilization in *Yki + dPGC1* tumors.

In sum, our findings demonstrate that *dPGC1* deregulation in cells with Yki activation leads to aberrant mitochondrial dynamics, increased Cyclin E levels, and DNA damage, factors that converge on key hallmarks of cancer, including altered cell proliferation, metabolism, and genomic instability. These results underscore the complexity of mitochondrial regulation in tumor biology and highlight the need to dissect the specific contributions of individual regulators of mitochondrial function. Targeting components of the mitochondrial fusion machinery may offer novel therapeutic strategies to limit Yki/YAP-driven tumor growth.

## Materials and methods

### *Drosophila* strains

The following stocks used in this paper were obtained from the Bloomington *Drosophila* Stock Center: *dPGC1¹* (synonym of *srl^KG08646*) (14965), *hsFLP, UAS-GFP.U;; tub-Gal4 FRT82B, tub-Gal80* (86311), *FRT82B* (2051), *FRT82B Wts^X1* (44251), *Gug-Gal4* (6773), *act-Gal4* (4414), *Mito-GFP* (8443), *yw* (1495), *UAS-LacZ* (3956), *UAS-dPGC1^RNAi* (33914), *EP-dPGC1* (20009), *UAS-dMfn* (67157), *UAS-dMfn^RNAi* (55189), *EP-Opa1* (20054), *UAS-Opa1^RNAi* (32358), *UAS-miro* (51646), and *UAS-dap^RNAi* (64026). The *UAS-dPGC1^shRNA* (v330271) and *UAS-miro^RNAi* (v106683) stocks were obtained from the Vienna *Drosophila* Stock Center. The *CycE^05306* stock was a kind gift from Helena Richardson. The following stocks used in this study are described in the following references: *MS1096-Gal4* [130], *ap-Gal4* [131], *UAS-Yki* [55], *UAS-InR29.4* (*UAS-InR*) [132], and *UAS-EGFR* [133].

*Drosophila* strains were raised on food that was made in-house containing agar (8 g/L), brewer's yeast (23.6 g/L), dextrose (50.8 g/L), and corn meal (58 g/L). Depending on the experiment, flies were housed in humidity-controlled incubators set at either 18 or 25°C with a 12/12-hour light–dark cycle.

## List of genotypes

### Fig 1

(A, B) *yw* corresponds to *yw/yw*.

*dPGC1¹* corresponds to *dPGC1¹/dPGC1¹*.

(C–E) Control corresponds to *MS1096-Gal4, UAS-RFP/+; +/+; UAS-LacZ/+. MS1096>dPGC1^{RNAi} corresponds to MS1096-Gal4, UAS-RFP/+; +/+; UAS-dPGC1^{RNAi}/+.*

*MS1096>dPGC1 corresponds to MS1096-Gal4, UAS-RFP/+; +/+; EP-dPGC1/+.*

## Fig 2

(A–E; J–Q) Control corresponds to *ap-Gal4, UAS-CD8-GFP/+; tub-Gal80ts/UAS-LacZ.*

*ap>dPGC1^{RNAi} corresponds to ap-Gal4, UAS-CD8-GFP/+; tub-Gal80ts/UAS-dPGC1^{RNAi}.*

*ap>Yki corresponds to ap-Gal4, UAS-CD8-GFP/+; UAS-Yki, tub-Gal80ts/UAS-LacZ.*

*ap>Yki+dPGC1^{RNAi} corresponds to ap-Gal4, UAS-CD8-GFP/+; UAS-Yki, tub-Gal80ts/UAS-dPGC1^{RNAi}.*

(F–I) Control corresponds to *hsFLP, UAS-GFP.U/+; +/+; tub-GAL4 FRT82B tub-GAL80/FRT82B Sb.*

*Wts^{X1} corresponds to hsFLP, UAS-GFP.U/+; +/+; tub-Gal4 FRT82B tub-Gal80/FRT82B Wts^{X1}.*

*Wts^{X1}+dPGC1^{shRNA} corresponds to hsFLP, UAS-GFP.U/+; UAS-dPGC1^{shRNA}/+; tub-Gal4 FRT82B tub-Gal80/FRT82B Wts^{X1}.*

## Fig 3

(A) The experimental condition corresponds to *ap-Gal4, UAS-CD8-GFP/+; UAS-Yki, tub-Gal80ts/UAS-dPGC1^{RNAi}* and the control used for normalization corresponds to *ap-Gal4, UAS-CD8-GFP/+; UAS-Yki, tub-Gal80ts/UAS-LacZ.*

(B) *ap-Gal4/+; UAS-Mito-GFP/+.*

(C) *Gug-Gal4, UAS-Mito-GFP/+.*

(D) *Gug-Gal4, UAS-Mito-GFP/UAS-dMfn.*

(E) *UAS-dMfn^{RNAi}/+; Gug-Gal4, UAS-Mito-GFP/+.*

(F) *EP-Opa1/+; Gug-Gal4, UAS-Mito-GFP/+.*

(G) *Gug-Gal4, UAS-Mito-GFP/UAS-Opa1^{RNAi}.*

(H) *UAS-miro/+; Gug-Gal4, UAS-Mito-GFP/+.*

(I) *UAS-miro^{RNAi}/+; Gug-Gal4, UAS-Mito-GFP/+.*

## Fig 4

*ap>Yki corresponds to ap-Gal4, UAS-CD8-GFP/+; UAS-Yki, tub-Gal80ts/UAS-LacZ.*

*ap>Yki+dPGC1^{RNAi} corresponds to ap-Gal4, UAS-CD8-GFP/+; UAS-Yki, tub-Gal80ts/UAS-dPGC1^{RNAi}.*

*ap>Yki+dPGC1^{RNAi}+dMfn^{RNAi} corresponds to ap-Gal4, UAS-CD8-GFP/UAS-dMfn^{RNAi}; UAS-Yki, tub-Gal80ts/UAS-dPGC1^{RNAi}.*

## Fig 5

**Control corresponds to** *ap-Gal4, UAS-CD8-GFP/+; tub-Gal80ts/UAS-Lac**Z**.*

*ap>Yki corresponds to ap-Gal4, UAS-CD8-GFP/+; UAS-Yki, tub-Gal80ts/UAS-LacZ.*

*ap>Yki+dPGC1^{RNAi}* corresponds to *ap-Gal4, UAS-CD8-GFP/+; UAS-Yki, tub-Gal80ts/UAS-dPGC1^{RNAi}.*

*ap>Yki+dPGC1^{RNAi}+dMfn^{RNAi}* corresponds to *ap-Gal4, UAS-CD8-GFP/UAS-dMfn^{RNAi}; UAS-Yki, tub-Gal80ts/ UAS-dPGC1^{RNAi}.*

*ap>Yki+dPGC1^{shRNA}* corresponds to *ap-Gal4, UAS-CD8-GFP/UAS-dPGC1^{shRNA}; UAS-Yki, tub-Gal80ts/+.*

*ap>Yki+dPGC1^{shRNA}+Opa1^{RNAi}* corresponds to *ap-Gal4, UAS-CD8-GFP/UAS-dPGC1^{shRNA}; UAS-Yki, tub-Gal80ts/ UAS-Opa1^{RNAi}.*

*ap>Yki+dMfn* corresponds to *ap-Gal4, UAS-CD8-GFP/+; UAS-Yki, tub-Gal80ts/UAS-dMfn.*

*ap>Yki+Opa1* corresponds to *ap-Gal4, UAS-CD8-GFP/EP-Opa1; UAS-Yki, tub-Gal80ts/+.*

**Fig 6**

**Control corresponds to** *ap-Gal4, UAS-CD8-GFP/+; tub-Gal80ts/UAS-LacZ.*

*ap>; CycE^{05306}* corresponds to *ap-Gal4, UAS-CD8-GFP/CycE^{05306}; tub-Gal80ts/+.*

*ap>Yki* corresponds to *ap-Gal4, UAS-CD8-GFP/+; UAS-Yki, tub-Gal80ts/UAS-LacZ.*

*ap>Yki+dap^{RNAi}* corresponds to *ap-Gal4, UAS-CD8-GFP/UAS-dap^{RNAi}; UAS-Yki, tub-Gal80ts/+.*

*ap>Yki+dPGC1^{RNAi}* corresponds to *ap-Gal4, UAS-CD8-GFP/+; UAS-Yki, tub-Gal80ts/UAS-dPGC1^{RNAi}.*

*ap>Yki+dPGC1^{RNAi}; CycE^{05306}* corresponds to *ap-Gal4, UAS-CD8-GFP/CycE^{05306}; UAS-Yki, tub-Gal80ts/UAS-dPGC1^{RNAi}.*

**S1 Fig**

*yw* corresponds to *yw/yw.*

*dPGC1^{1}* corresponds to *dPGC1^{1}/dPGC1^{1}.*

**S2 Fig**

*act>* corresponds to *act-Gal4/+.*

*act>dPGC1^{RNAi}* corresponds to *act-Gal4/UAS-dPGC1^{RNAi}.*

*act>dPGC1^{shRNA}* corresponds to *UAS-dPGC1^{shRNA}/+; act-Gal4/+.*

*act>dMfn* corresponds to *act-Gal4/UAS-dMfn.*

*act>dMfn^{RNAi}* corresponds to *UAS-dMfn^{RNAi}/+; act-Gal4/+.*

*act>Opa1* corresponds to *EP-Opa1/+; act-Gal4/+.*

*act>Opa1^{RNAi}* corresponds to *act-Gal4/UAS-Opa1^{RNAi}.*

*act>miro* corresponds to *UAS-miro/+; act-Gal4/+.*

*act>miro^{RNAi}* corresponds to *UAS-miro^{RNAi}/+; act-Gal4/+.*

**S3 Fig**

Control corresponds to *ap-Gal4, UAS-CD8-GFP/+; tub-Gal80ts/UAS-LacZ.*

*ap>dPGC1^{RNAi}* corresponds to *ap-Gal4, UAS-CD8-GFP/+; tub-Gal80ts/UAS-dPGC1^{RNAi}.*

**S4 Fig**

(A–C) *ap>EGFR* corresponds to *ap-Gal4, UAS-CD8-GFP/+; UAS-EGFR, tub-Gal80ts/+.*

*ap>EGFR+dPGC1$^{RNAi}$* corresponds to *ap-Gal4, UAS-CD8-GFP/+; UAS-EGFR, tub-Gal80ts/UAS-dPGC1$^{RNAi}$.*

(D–F) *ap>InR* corresponds to *ap-Gal4, tub-Gal80ts/+; UAS-InR/+.*

*ap>InR+dPGC1$^{RNAi}$* corresponds to *ap-Gal4, tub-Gal80ts/+; UAS-InR/UAS-dPGC1$^{RNAi}$.*

**S5 Fig**

Control corresponds to *ap-Gal4, UAS-CD8-GFP/+; tub-Gal80ts/UAS-LacZ.*

*ap>dPGC1$^{shRNA}$* corresponds to *ap-Gal4, UAS-CD8-GFP/ UAS-dPGC1$^{shRNA}$; tub-Gal80ts/+.*

*ap>Yki* corresponds to *ap-Gal4, UAS-CD8-GFP/+; UAS-Yki, tub-Gal80ts/UAS-LacZ .*

*ap>Yki+dPGC1$^{shRNA}$* corresponds to *ap-Gal4, UAS-CD8-GFP/UAS-dPGC1$^{shRNA}$; UAS-Yki, tub-Gal80ts/+.*

**S6 Fig**

*ap>Yki* corresponds to *ap-Gal4, UAS-CD8-GFP/+; UAS-Yki, tub-Gal80ts/UAS-LacZ.*

*ap>Yki+dPGC1$^{RNAi}$* corresponds to *ap-Gal4, UAS-CD8-GFP/+; UAS-Yki, tub-Gal80ts/UAS-dPGC1$^{RNAi}$.*

**S7 Fig**

The experimental condition corresponds to *ap-Gal4, UAS-CD8-GFP/+; tub-Gal80ts/UAS-dPGC1$^{RNAi}$* and the control used for normalization corresponds to *ap-Gal4, UAS-CD8-GFP/+; tub-Gal80ts/UAS-LacZ.*

**S8 Fig**

*ap>Yki* corresponds to *ap-Gal4, UAS-CD8-GFP/+; UAS-Yki, tub-Gal80ts/UAS-LacZ.*

*ap>Yki+dPGC1* corresponds to *ap-Gal4, UAS-CD8-GFP/+; UAS-Yki, tub-Gal80ts/EP-dPGC1.*

**S9 Fig**

Control corresponds to *Gug-Gal4, UAS-Mito-GFP/+.*

*Gug>dMfn* corresponds to *Gug-Gal4, UAS-Mito-GFP/UAS-dMfn.*

*Gug>Mfn$^{RNAi}$* corresponds to *UAS-dMfn$^{RNAi}$/+; Gug-Gal4, UAS-Mito-GFP/+.*

*Gug>Opa1* corresponds to *EP-Opa1/+; Gug-Gal4, UAS-Mito-GFP/+.*

*Gug>Opa1$^{RNAi}$* corresponds to *Gug-Gal4, UAS-Mito-GFP/UAS-Opa1$^{RNAi}$.*

*Gug>miro* corresponds to *UAS-miro/+; Gug-Gal4, UAS-Mito-GFP/+.*

*Gug>miro$^{RNAi}$* corresponds to *UAS-miro$^{RNAi}$/+; Gug-Gal4, UAS-Mito-GFP/+.*

**S10 Fig**

Control corresponds to *ap-Gal4, UAS-CD8-GFP/+; tub-Gal80ts/UAS-LacZ.*

*ap>Mfn* corresponds to *ap-Gal4, UAS-CD8-GFP/+; tub-Gal80ts/ UAS-dMfn.*

*ap>Mfn$^{RNAi}$* corresponds to *ap-Gal4, UAS-CD8-GFP/ UAS-dMfn$^{RNAi}$; tub-Gal80ts/+.*

ap>Opa1 corresponds to *ap-Gal4, UAS-CD8-GFP/EP-Opa1; tub-Gal80ts/+*.

ap>Opa1<sup>RNAi</sup> corresponds to *ap-Gal4, UAS-CD8-GFP/+; tub-Gal80ts/ UAS-Opa1<sup>RNAi</sup>*.

**S11 Fig**

(A–C) *ap>Yki+dPGC1<sup>RNAi</sup>* corresponds to *ap-Gal4, UAS-CD8-GFP/+; UAS-Yki, tub-Gal80ts/UAS-dPGC1<sup>RNAi</sup>*.

*ap>Yki+dPGC1<sup>RNAi</sup>+miro<sup>RNAi</sup>* corresponds to *ap-Gal4, UAS-CD8-GFP/UAS-miro<sup>RNAi</sup>; UAS-Yki, tub-Gal80ts/ UAS-dPGC1<sup>RNAi</sup>*.

(D–F) *ap>Yki* corresponds to *ap-Gal4, UAS-CD8-GFP/+; UAS-Yki, tub-Gal80ts/UAS-LacZ*.

*ap>Yki+miro* corresponds to *ap-Gal4, UAS-CD8-GFP/UAS-miro; UAS-Yki, tub-Gal80ts/+*.

**S12 Fig**

*ap>Yki* corresponds to *ap-Gal4, UAS-CD8-GFP/+; UAS-Yki, tub-Gal80ts/UAS-LacZ*.

*ap>Yki+dPGC1<sup>RNAi</sup>* corresponds to *ap-Gal4, UAS-CD8-GFP/+; UAS-Yki, tub-Gal80ts/UAS-dPGC1<sup>RNAi</sup>*.

**S13 Fig**

*ap>Yki* **corresponds to***ap-Gal4, UAS-CD8-GFP/+; UAS-Yki, tub-Gal80ts/UAS-LacZ*.

*ap>Yki+dap<sup>RNAi</sup>* corresponds to *ap-Gal4, UAS-CD8-GFP/UAS-dap<sup>RNAi</sup>; UAS-Yki, tub-Gal80ts/+*.

**S14 Fig**

*yw* corresponds to *yw/yw.*

*dPGC1<sup>1</sup>* corresponds to *dPGC1<sup>1</sup>/dPGC1<sup>1</sup>*.

### Temporal control of transgene expression by Gal80ts

For experiments using ap-Gal4 or act-Gal4, the Gal4/Gal80ts system was used to prevent lethality during early development. This system allowed UAS-driven transgenes to be controlled in a temperature-dependent manner. *Drosophila* crosses were kept for 2 days at 18°C to lay eggs. After 2 days, crosses were flipped into new vials and the old vials containing eggs were kept for 2 additional days at 18°C before being transferred to a 29°C incubator for transgene expression to be induced. To ensure that wing discs were analyzed at the same developmental stage, larvae with hyperplastic tumors were dissected after 5 days at 29°C, whereas giant larvae containing neoplastic tumors were dissected after 7 days at 29°C.

### Induction of mitotic clones

Mitotic recombination clones were generated using the FLP/FRT system, employing the MARCM (Mosaic Analysis with a Repressible Cell Marker) technique. Larvae carrying the necessary transgenes (see the list of genotypes for details) were maintained at 25°C for 48–96 hours after egg laying. To induce FLP recombinase expression, vials were heat-shocked by immersion in a circulating water bath at 37°C for 60 min. Following heat shock, larvae were returned to 25°C and allowed to develop for an additional 96 hours to permit clone growth. Larvae were dissected at the third instar wandering stage, corresponding to 96 hours after heat shock, for analysis of imaginal discs.

### Immunohistochemistry

Larvae were dissected in PBS and fixed in 3.7% formaldehyde/PBS for 20 min at room temperature. Samples were then washed in 0.2% Triton/PBS (PBT) and blocked for an hour at room temperature in blocking buffer (BBT) made of 0.2%

Triton, 5mM NaCl, and 3% BSA diluted in PBS. Primary antibodies were diluted in BBT, and the samples were left in primary antibody solution overnight at room temperature. Longer primary antibody incubations were done at 4°C. After removing the primary antibody and performing washes with BBT, samples were incubated for 2 hours at room temperature with the relevant fluorescent secondary antibodies and DAPI diluted in BBT. After removing the secondary antibody and performing washes with PBT, samples were mounted in 90% glycerol/ PBS containing 0.5% N-propyl gallate. Samples were imaged with the Leica SP8 confocal microscope. For stainings intended for mitochondrial analysis, BBT and PBT contained 0.2% Tween instead of 0.2% Triton.

The following primary antibodies were used: mouse anti-Mmp1 (Developmental Studies Hybridoma Bank, 3A6B4/5H7B11/3B8D12 mixed in equal volumes), rabbit anti-pH2Av (Rockland, 600-401-914, dilution 1:1000), rabbit anti-PH3 (Cell Signal Technology, 9701, dilution 1:100), and rabbit anti-Cyclin E (Santa Cruz, 33748, dilution 1:100). The following secondary antibodies were used: Alexa Fluor 635-phalloidin to stain F-actin (Invitrogen, A34054, dilution 1:400), anti-mouse Alexa Fluor 555 (Invitrogen, A21425, dilution 1:400), and anti-rabbit Alexa Fluor 555 (Invitrogen, A21430, dilution 1:400). DAPI (Invitrogen, D1306) was used at a 600 nM concentration to stain the nuclei.

Due to the absence of a specific antibody against dPGC1, all conclusions regarding its role are based on transcript and genetic evidence.

### Analysis of mitochondrial membrane potential

Wing discs from *Drosophila* larvae were dissected in Schneider's medium (Sigma, Ref: S9895). After dissection, wing discs were incubated in 100 nM TMRE (Sigma, Ref: 87917, dissolved in Schneider's medium) for 20 min with agitation. Then, wing discs were washed once with Schneider's medium for 5 min, rinsed with PBS, and mounted in Schneider's medium. Samples were immediately imaged after mounting with the Leica SP8 confocal microscope. All steps were performed at room temperature.

### Electron microscopy

Larval wing discs were dissected in PBS and fixed in 2% glutaraldehyde/0.1M sodium cacodylate buffer at room temperature. Sample preparation was performed by the Core Facility for Integrative Microscopy at the University of Copenhagen, where tissues were fixed, embedded, stained, dehydrated, infiltrated, embedded, and cut into thin sections for imaging. All specimens were imaged at the Core Facility for Integrative Microscopy with a Transmission Electron Microscope (Philips CM100, FEL). Digital images were recorded with an OSIS Veleta digital slow-scan 2k × 2k CCD camera and the ITEM software package. Mitochondrial morphologies reflected in these 2D micrographs were analyzed with Fiji software. Mitochondria were manually outlined and quantified for area, perimeter, and aspect ratio (calculated as MaxFeret/MinFeret to assess elongation). At least 164 mitochondria were quantified per genotype.

### RNA extraction, cDNA synthesis, and qPCR

For experiments performed with ap-Gal4, RNA was extracted from wing discs dissected from third instar wandering larvae with the TRIzol reagent (Life Technologies, Ref: 15596026). For experiments performed with act-Gal4, RNA was extracted from whole third instar wandering larvae with the RNeasy Mini Kit (Qiagen, Ref: 74106). Total RNA was treated with RQ1 DNase I (Promega, Ref: M6101) and converted to cDNA using the SuperScript III Reverse Transcriptase kit (Life Technologies, Ref: 18080-044). qPCR was performed using 5× HOT FIREPol EvaGreen qPCR Mix Plus (Solis Biodyne, Ref: 08-24-00001) on the QuantStudio 6 Flex Real-Time PCR machine (Applied Biosystems). The primers used are listed below.

| Gene | Forward primer sequence: | Reverse primer sequence: |
|------|--------------------------|--------------------------|
| dPGC1 | GCTGACTTGGATAATGCTGGCA | TGAACCTACCGGTGACTTGC |
| dERR | GGAATTGGTCAGCGTCATTGG | GAACTCCGTGTAACCGCACT |
| Ets97D | GGTGCCCTGGAAGAAGTGATT | ACGCATTGGTCCACGAGATT |
| ewg | CGATGTTGATTACACCACGCA | TTTGAATGGTAGCACCGCCT |
| dMfn | AGCAGTATTGGACACAGCGT | ACGCCAGCCGATAGTTTTCA |
| Opa1 | CCCATGAAGGCTTTGGGCTA | ACCCCTCGACGATGGAAAAG |
| Drp1 | AGATGCTGCGCTTTCCCAAA | AGTAGGGATCGCTGTCCGTT |
| Pink1 | TACCGACAGGACCAATTGCC | CCACTGTAGGATCTCCGGACT |
| Park | CGGATGTGAGTGATACCGTGT | TACTCCTCGCGTGTCAACAG |
| Atg7 | GTGACACGACCCGGAGTATC | ATTTAGGACGGCCGCTGAAC |
| milt | AAGGACCTAGAGCTGACCGT | GGAGTGTCAGTATCTGTGCCA |
| miro | TATTCCACCATGGACAGCGT | CGGATGTGAGCTCCTGTTCTTC |
| CycE | TTCTACCAGCAGCACTGAGC | TGGAAGGATAGCGATTGGGC |
| Yki | GATCTGGCCATCATTGGTCT | GTCATCAGCCCCATTCAGTT |
| Diap1 | ATGAGAGTGATGTCTGCTGCTCTTC | AGCTGATGAAGGGTCAGCTCTATATC |
| Ban | ACCGGTTTTCGATTTGGTTT | ACCATCGGAATGTGGAATGT |
| Dap | TGGCAATGCGGCTTAGCTC | GCTTTGCGGGCGATAGTTTCT |
| RP49 | AAGCGGCGACGCACTCTGTT | GCCCAGCATACAGGCCCAAG |

## Western blot

Wing discs were dissected from *Drosophila* larvae and transferred into RIPA protein extraction and immunoprecipitation buffer (Sigma-Aldrich, Ref: R0278) containing cOmplete Protease Inhibitor Cocktail (Merck, Ref: 04693124001), PhosSTOP (Merck, Ref: 04906837001), and 100mM PMSF (Sigma-Aldrich, Ref: 93482). Wing discs were homogenized by pipetting up and down and then leaving samples on ice for 30 min. Afterwards, samples were sonicated three times for three cycles with the following settings: 100% amplitude, 1 cycle. Samples were then centrifuged for 20 min at 15,000$g$ and 4°C to obtain the supernatant that was used for western blots. Protein concentration was quantified using the Pierce BCA protein assay kit (Thermo Fisher Scientific, Ref: 23227). Western blots were performed using 15 µg of protein per sample. Western blot membranes were blocked for an hour at room temperature with 5% skimmed milk (Fluka, Ref: 70166-500G) dissolved in TBST. After washing with TBST, the membranes were incubated overnight at 4°C with primary antibodies diluted in TBST. After removing the primary antibody and performing washes with TBST, samples were incubated for 1 hour at room temperature with the relevant HRP-conjugated secondary antibodies. After removing the secondary antibodies and washing with TBST, the Pierce ECL Western Blotting Substrate kit (Thermo Fisher Scientific, Ref: 32106) was used to visualize protein levels on the Amersham Imager 600 (GE Healthcare).

The following primary antibodies were used: rabbit anti-Cyclin E (Santa Cruz, 33748, dilution 1:1000) and mouse anti-α-Tubulin (Developmental Studies Hybridoma Bank, 12G10, dilution 1:1000). The following HRP-conjugated secondary antibodies were used: polyclonal goat anti-rabbit-HRP (Invitrogen, 31460, dilution 1:10000) and polyclonal goat anti-mouse-HRP (Dako, P0447, dilution 1:5000). Note that since Cyclin E and α-Tubulin have similar molecular weights, membranes had to be stripped to detect both proteins without any crossover.

## Image analysis and processing

Image analyses were performed using Fiji (ImageJ) software. Specific quantification procedures are described in the sections below. Final figures were assembled using Adobe Photoshop.

## Quantification of *Drosophila* wing imaginal disc and clone size

To quantify wing imaginal disc size, the "threshold" function was first used to select either the GFP-positive area or the DAPI-positive area, depending on the experiment. After setting a threshold, areas were determined using the "analyze particles" option. A minimum of 10 discs were quantified per genotype. To quantify clone size, individual clones in each wing disc were selected with the "polygon" tool, and their area was directly measured using the "measure" option. For each genotype, at least 65 clones were quantified from a total of a minimum of 10 discs. For both wing disc and clone size quantifications, results were expressed as the fold change of GFP or DAPI area relative to the corresponding control.

## Quantification of *Drosophila* adult wing size

Crosses were allowed to lay eggs on apple agar plates with yeast paste at 25°C for 10–14 hours. Once eggs hatched and L1 larvae emerged, 20 larvae were transferred to vials containing standard *Drosophila* food, and these vials were kept at 25°C until adult flies eclosed. Wings were fixed by introducing adult flies into a 25% glycerol/75% ethanol mixture for at least 24 hours. After that, wings were plucked and mounted in glycerol for imaging. Images of adult *Drosophila* wings were taken with a Leica M165 stereomicroscope.

To measure adult wing area, the "polygon" selection tool in Fiji was used to draw a border around the wing edges. Next, the "measure" option was used to quantify the wing area. Results were expressed as the fold change in wing area relative to the control. A total of 20 wings were quantified per genotype.

## Quantification of mitochondrial shape in the peripodial membrane

The Gug-Gal4 driver was used to drive the expression of the Mito-GFP mitochondrial marker in the peripodial membrane. After obtaining images of peripodial membrane cells, two filters were applied to the GFP channel, unsharp mask (radius = 10.0 pixels, mask strength = 0.9) and median filtering (radius = 3), to enhance and smooth the Mito-GFP signal, respectively [134]. Next, the "threshold" function was used to create a binary mask, and the "Mitochondrial Analyzer 2D Analysis" plugin from ImageJ [135] was used to calculate the mean form factor. Mitochondria from a minimum of 15 images were quantified per genotype.

## Quantification of TMRE intensity

For each image, the area of the wing disc containing mitochondria stained with TMRE was selected using the "threshold" function on the TMRE channel. The mean gray value of only the thresholded selection was obtained with the "measure" function. Results were expressed as the fold change of TMRE intensity relative to the control. Importantly, comparisons were only made between wing discs that had been stained and imaged together, meaning that fold changes were always calculated considering the mean of the control for each individual experiment and then all fold changes were pooled for the statistical analysis. For each genotype, a minimum of 23 wing discs from a total of 5 experiments were quantified.

## Quantification of Mmp1 area

For each wing disc, the GFP-positive area was first measured as explained above and the GFP selection was added to the ROI manager. Afterwards, a "gaussian blur" filter of 1 was applied to the Mmp1 channel and the "threshold" function was used to select only the Mmp1-positive signal. The thresholded Mmp1 area within the previously selected GFP area was measured with the "measure" option within the ROI manager. Results were expressed as the Mmp1 area normalized to the GFP-positive area. A minimum of 11 wing discs were quantified per genotype.

### Quantification of F-actin and Cyclin E intensity

Quantification of F-actin and Cyclin E intensities were done with the same procedure. The "polygon" tool was first used to select a DAPI-positive and GFP-positive (if applicable) area within the image. Then, the mean gray value of the corresponding channel (either F-actin or Cyclin E) within the selected area was obtained with the "measure" function. Results were expressed as the direct values of mean intensity (for F-actin) or as the fold change in mean intensity relative to the control (for Cyclin E). A minimum of 19 wing discs (for F-actin) or 13 wing discs (for Cyclin E) were quantified per genotype.

### Quantification of PH3-positive cells

First, the "polygon" tool was used to select a DAPI-positive and GFP-positive area, which was then measured with the "measure" function. Then, a "gaussian blur" filter of 3 was applied to the PH3 channel. Finally, the number of PH3-positive cells was measured with the "find maxima" tool. Results were expressed as the number of PH3-positive cells per GFP-positive area. A minimum of 19 wing discs were quantified per genotype.

### Quantification of pH2Av-positive foci

Maximum projections of Z-stack images of wing discs were used for this quantification. First, the GFP-positive area was measured as previously described. Then, the number of pH2Av-positive foci was measured using the "3D Objects Counter". Results were expressed as the fold change in number of pH2Av-positive foci per area relative to the control. A minimum of 18 wing discs were quantified per genotype.

### Quantification of protein levels in western blot membranes

Western blot membranes were loaded to Fiji, and the "rectangle" tool was used to select the corresponding bands, fitting all genotypes in the same rectangle. Within the "gels" function, "select first lane" was used to set this rectangle as a lane, and then "plot lanes" was used to generate a lane profile plot showcasing intensity peaks corresponding to each band within the rectangle. The "line" tool was used to draw baselines for each peak, generating closed areas that did not include the background signal, and these peak areas were then measured. For each genotype, areas obtained for Cyclin E bands were normalized to the corresponding α-Tubulin bands. Results were expressed as the fold change in Cyclin E/α-Tubulin peak areas for each membrane, and then all fold changes were pooled for the statistical analysis. A total of 7 western blot membranes were quantified.

### Statistics

Graphs and statistical analyses were done using GraphPad Prism 10. Data is always presented as full-range box plots. The number of biological replicates quantified per sample and the statistical tests applied to determine statistical significance are indicated in the corresponding figure legends.

### Supporting information

**S1 Fig. *dPGC1* mutant wing discs. (A, B)** Confocal micrographs of *Drosophila* wing imaginal discs from the following genotypes: *yw/yw* (control, A) and *dPGC1¹/dPGC1¹* (mutant, B). DAPI labels the DNA and is shown in grayscale. Scale bars, 100 μm. **(C)** Quantification of wing disc area of the genotypes in A and B. Wing disc size was normalized to the mean area of the control (*yw/yw*). Statistical significance was determined using an unpaired $t$ test ($n = 16$ [*yw*], $n = 13$ [*dPGC1¹*]). **$p < 0.01$. The data underlying this graph can be found in S2 Data.
(PDF)

**S2 Fig. UAS-driven transgene efficiency.** mRNA quantification by qPCR of the transgenes used to modulate the genes *dPGC1* **(A)**, *dMfn* **(B)**, *Opa1* **(C)**, and *miro* **(D)**. The transgenes were expressed under the control of the act-Gal4 driver and whole larvae were used for the analysis. The control genotype is *act-Gal4/+*. Each genotype was run in three to five biological replicates. RP49 was used as the housekeeping gene. Statistical significance was determined using unpaired *t* tests with or without Welch's correction depending on whether variances were significantly different or not, respectively. *$p < 0.05$, **$p < 0.01$, ***$p < 0.001$. Additional information about these transgenes is shown in S1 Data. The data underlying all the graphs in the figure can be found in S2 Data.
(PDF)

**S3 Fig. *Yki* expression and activity in discs with reduced dPGC1.** mRNA quantification by qPCR of *Yki* **(A)** and the Yki target genes *CycE* **(B)**, *Diap1* **(C)**, and *Ban* **(D)** in *ap-Gal4, UAS-GFP, UAS-dPGC1-RNAi* wing imaginal discs. The control genotype is *ap-Gal4, UAS-GFP, UAS-LacZ*. Each genotype was run in four biological replicates. RP49 was used as the housekeeping gene. Statistical significance was determined using unpaired *t* tests. ns, non-significant. The data underlying all the graphs in the figure can be found in S2 Data.
(PDF)

**S4 Fig. *dPGC1* depletion does not cooperate with the oncogenes *EGFR* and *InR*. (A, B)** Confocal micrographs of wing imaginal discs of the following genotypes: *ap-Gal4, UAS-EGFR, UAS-GFP, UAS-LacZ* (A) and *ap-Gal4, UAS-EGFR, UAS-GFP, UAS-dPGC1-RNAi* (B). GFP is shown in green. DAPI labels the DNA and is shown in blue. Scale bars, 100 µm. **(C)** Quantification of wing disc area (GFP-positive area) of the genotypes in A and B. GFP-positive area was normalized to the mean of the control (*ap>EGFR*). Statistical significance was determined using an unpaired *t* test ($n = 11$ [*ap>EGFR*], $n = 11$ [*ap>EGFR, dPGC1-RNAi*]). *$p < 0.05$. **(D, E)** Confocal micrographs of wing imaginal discs of the following genotypes: *ap-Gal4, UAS-InR, UAS-GFP* (GFP not shown) (D) and *ap-Gal4, UAS-InR, UAS-dPGC1-RNAi* (E). DAPI labels the DNA and is shown in grayscale. Scale bars, 100 µm. **(F)** Quantification of wing disc area (DAPI-positive area) of the genotypes in D and E. DAPI-positive area was normalized to the mean of the control (*ap>InR*). Statistical significance was determined using an unpaired *t* test ($n = 19$ [*ap>InR*], $n = 11$ [*ap>InR, dPGC1-RNAi*]). **$p < 0.01$. The data underlying all the graphs in the figure can be found in S2 Data.
(PDF)

**S5 Fig. Oncogenic cooperation between Yki and dPGC1. (A, B)** Confocal micrograph showing an *ap-Gal4, UAS-GFP, UAS-Yki, UAS-dPGC1-shRNA* third instar wing imaginal disc. F-actin labels cell polarity and is shown in grayscale. Mmp1 is shown in red. GFP is shown in green. DAPI labels the DNA and is shown in blue. The dashed white box in A indicates the region of the wing disc that is shown as a magnification in B. Scale bars, 100 µm. **(C)** Quantification of GFP-positive area in third instar wing imaginal discs of the indicated genotypes. GFP-positive areas were normalized to the mean of the control (*ap > LacZ*). Statistical significance was determined using unpaired *t* tests with Welch's correction ($n = 10$ [Control], $n = 10$ [*ap > dPGC1-shRNA*], $n = 10$ [*ap > Yki*], $n = 10$ [*ap > Yki, dPGC1-shRNA*]). **$p < 0.01$, ****$p < 0.0001$. The data underlying this graph can be found in S2 Data.
(PDF)

**S6 Fig. Membrane Potential in *Yki + dPGC1-RNAi* tumors. (A, B)** Confocal micrographs showing magnifications from tumorous wing discs with the following genotypes: *ap-Gal4, UAS-Yki, UAS-GFP, UAS-LacZ* (A) and *ap-Gal4, UAS-Yki, UAS-GFP, UAS-dPGC1-RNAi* (B). GFP is shown in green. TMRE staining is shown in red. Scale bars, 10 µm. **(C)** Quantification of TMRE intensity in the genotypes in A and B. TMRE intensity was normalized to the mean of the control (*ap > Yki*). Statistical significance was determined using a Mann-Whitney test for non-parametric data ($n = 23$ [*ap > Yki*], $n = 56$ [*ap > Yki, dPGC1-RNAi*]). ns, non-significant. The data underlying this graph can be found in S2 Data.
(PDF)

**S7 Fig. Expression of genes controlling mitochondrial dynamics in discs with reduced dPGC1.** mRNA quantification by qPCR of the indicated genes in *ap-Gal4, UAS-GFP, UAS-dPGC1-RNAi* wing imaginal discs using *ap-Gal4, UAS-GFP, UAS-LacZ* as the control genotype. Genes are separated in different categories: mitochondria biogenesis (blue), mitochondria dynamics (black), mitophagy (pink), and mitochondria transport (green). Each genotype was run in four biological replicates. RP49 was used as the housekeeping gene. Statistical significance was determined using unpaired *t* tests with Welch's correction for parametric data and Mann–Whitney tests for non-parametric data. *$p < 0.05$, ***$p < 0.001$. The data underlying this graph can be found in S2 Data.
(PDF)

**S8 Fig. *dPGC1* overexpression in a context of *Yki* upregulation. (A, B)** Confocal micrographs of wing imaginal discs of the following genotypes: *ap-Gal4, UAS-Yki, UAS-GFP, UAS-LacZ* (A) and *ap-Gal4, UAS-Yki, UAS-GFP, EP-dPGC1* (B). GFP is shown in green. DAPI labels the DNA and is shown in blue. Scale bars, 100 μm. **(C)** Quantification of GFP-positive area in third instar wing imaginal discs of the genotypes in A and B. GFP-positive areas were normalized to the mean of the control (*ap > Yki*). Statistical significance was determined using an unpaired *t* test ($n = 14$ [*ap > Yki*], $n = 10$ [*ap > Yki, dPGC1*]). ns, non-significant. **(D, E)** mRNA quantification by qPCR of *dMfn* (D) and *Opa1* (E) in the indicated genotypes. The control genotype is *ap > Yki*. Each genotype was run in five biological replicates. RP49 was used as the housekeeping gene. Statistical significance was determined using unpaired *t* tests. ns, non-significant. The data underlying all the graphs in the figure can be found in S2 Data.
(PDF)

**S9 Fig. Changes in mitochondrial shape upon *dMfn*, *Opa1*, and *miro* gene manipulation.** Quantification of mitochondrial mean form factor (shape measure where a value of 1 indicates a round object and values increase with elongation) from confocal micrographs obtained from the peripodial membrane of the following genotypes: *Gug-Gal4, UAS-Mito-GFP* (Control)*; Gug-Gal4, UAS-Mito-GFP, UAS-dMfn*; *Gug-Gal4, UAS-Mito-GFP, UAS-dMfn-RNAi*; *Gug-Gal4, UAS-Mito-GFP, EP-Opa1; Gug-Gal4, UAS-Mito-GFP, UAS-Opa1-RNAi; Gug-Gal4, UAS-Mito-GFP, UAS-miro*; and *Gug-Gal4, UAS-Mito-GFP, UAS-miro-RNAi.* Statistical significance was determined using unpaired *t* tests with Welch's correction for parametric data and Mann–Whitney tests for non-parametric data ($n = 20$ [Control], $n = 20$ [*Gug > dMfn*], $n = 20$ [*Gug > dMfn-RNAi*], $n = 23$ [*Gug > Opa1*], $n = 23$ [*Gug > Opa1-RNAi*], $n = 20$ [*Gug>miro*], $n = 20$ [*Gug>miro-RNAi*]). ** $p < 0.01$, ***$p < 0.001$, ****$p < 0.0001$. The data underlying this graph can be found in S2 Data.
(PDF)

**S10 Fig. Wing disc size upon *dMfn* and *Opa1* manipulation. (A–E)** Confocal micrographs showing discs of the following genotypes: *ap-Gal4, UAS-GFP, UAS-LacZ* (A); *ap-Gal4, UAS-GFP, UAS-dMfn* (B); *ap-Gal4, UAS-GFP, UAS-dMfn-RNAi* (C); *ap-Gal4, UAS-GFP, EP-Opa1* (D); and *ap-Gal4, UAS-GFP, UAS-Opa1-RNAi* (E). GFP is shown in green. DAPI labels the DNA and is shown in blue. Scale bars, 100 μm. **(F)** Quantification of GFP-positive area in third instar wing imaginal discs of the genotypes shown in A–E. GFP-positive area was normalized to the mean of the control (*ap > LacZ*). Statistical significance was determined using unpaired *t* tests with Welch's correction for parametric data and Mann–Whitney tests for non-parametric data ($n = 18$ [Control], $n = 18$ [*ap > dMfn*], $n = 20$ [*ap > dMfn-RNAi*], $n = 20$ [*ap > Opa1*], $n = 19$ [*ap > Opa1-RNAi*]). *$p < 0.05$. The data underlying this graph can be found in S2 Data.
(PDF)

**S11 Fig. Miro does not affect Yki-driven tumors. (A, B)** Confocal micrographs of wing imaginal discs of the following genotypes: *ap-Gal4, UAS-Yki, UAS-GFP, UAS-dPGC1-RNAi* (A) and *ap-Gal4, UAS-Yki, UAS-GFP, UAS-dPGC1-RNAi, UAS-miro-RNAi* (B). GFP is shown in green. DAPI labels the DNA and is shown in blue. Scale bars, 100 μm. **(C)** Quantification of GFP-positive area in wing imaginal discs of the genotypes in A and B. GFP-positive areas were normalized to the

mean of the control (*ap > Yki, dPGC1-RNAi*). Statistical significance was determined using a Mann–Whitney test (*n* = 22 [*ap > Yki, dPGC1-RNAi*], *n* = 26 [*ap > Yki, dPGC1-RNAi, miro-RNAi*]). ns, non-significant. **(D, E)** Confocal micrographs of wing imaginal discs of the following genotypes: *ap-Gal4, UAS-Yki, UAS-GFP, UAS-LacZ* (D) and *ap-Gal4, UAS-Yki, UAS-GFP, UAS-miro* (E). GFP is shown in green. DAPI labels the DNA and is shown in blue. Scale bars, 100 µm. **(F)** Quantification of GFP-positive area in wing imaginal discs of the genotypes in D and E. GFP-positive areas were normalized to the mean of the control (*ap > Yki*). Statistical significance was determined using an unpaired *t* test (*n* = 21 [*ap > Yki*], *n* = 25 [*ap > Yki, miro*]). ns, non-significant. The data underlying all the graphs in the figure can be found in S2 Data.
(PDF)

**S12 Fig. Representative whole western membrane images of Cyclin E protein levels in *Yki + dPGC1-RNAi* tumors. (A–D)** Examples of two western blot membranes to analyze protein levels of Cyclin E (A and C) and α-Tubulin (B and D) in wing imaginal discs of the following genotypes: *ap-Gal4, UAS-Yki, UAS-GFP, UAS*-LacZ; and *ap-Gal4, UAS-Yki, UAS-GFP, UAS-dPGC1-RNAi.* The molecular weights (in kDa) of the visible bands of the ladder are indicated at the left of each membrane. The dashed red boxes in A–D indicate the regions of the membranes that are shown as magnifications in A′–D′, respectively. Note that the western blot membrane of panels A and B corresponds to the one shown in Fig 6B and therefore panels A′ and B′ are the same as those in Fig 6B. The membranes shown here include additional genotypes not relevant to this study. Only the lanes corresponding to the relevant samples are indicated.
(PDF)

**S13 Fig. *UAS-Dap-RNAi* transgene efficiency.** mRNA quantification by qPCR of *dap* in wing imaginal discs of the following genotypes: *ap-Gal4, UAS-Yki, UAS-GFP, UAS-LacZ* and *ap-Gal4, UAS-Yki, UAS-GFP, UAS-dap-RNAi*. The control genotype is *ap > Yki.* Each genotype was run in five biological replicates. RP49 was used as the housekeeping gene. Statistical significance was determined using an unpaired *t* test. ****$p < 0.0001$. The data underlying this graph can be found in S2 Data.
(PDF)

**S14 Fig. Cyclin E in wing imaginal discs of *dPGC1* mutant flies. (A, B)** Confocal micrographs showing the wing pouch of imaginal discs of the following genotypes: *yw/yw* (control, A) and *dPGC1¹/dPGC1¹* (mutant, B). Cyclin E is shown in red. DAPI labels the DNA and is shown in blue. Scale bars, 10 µm. **(C)** Quantification of Cyclin E protein mean intensity of the genotypes in A and B. Cyclin E intensity was normalized to the mean intensity of the control (*yw/yw*). Statistical significance was determined using an unpaired *t* test (*n* = 22 [*yw*], *n* = 13 [*dPGC1¹*]). ns, non-significant. The data underlying this graph can be found in S2 Data.
(PDF)

**S1 Data. Genetic constructs modulating mitochondrial function in *Drosophila* and their efficiency under actin-Gal4.** It includes: (1) Construct name and target gene (e.g., UAS-dPGC1-RNAi, UAS-Mfn, EP-Opa1), (2) Chromosomal insertion site, (3) Efficiency of expression or phenotypic impact, reported as fold change over control ± standard deviation, and (4) Stock identifiers from BDSC or VDRC.
(XLSX)

**S2 Data. Source data.** Numerical data used to generate the graphs in each of the figures, shown in separate tabs.
(XLSX)

**S1 Raw Images. Original blot and gel images corresponding to Figs 6B and S12 of the manuscript.** These images show the original western blot membranes used to analyze Cyclin E and α-Tubulin protein levels in wing imaginal discs from *Drosophila* genotypes with *Yki* overexpression and *dPGC1* depletion. Molecular weight markers are indicated, and the relevant lanes are annotated to match the experimental conditions described in the manuscript. Lanes not relevant to

this study have been labeled as "X" to clearly distinguish them from the experimental samples. These raw data images are provided in compliance with PLOS Biology guidelines for transparency and reproducibility.
(PDF)

## Acknowledgments

We thank Helena Richardson, Kim F. Rewitz, and Takashi Koyama for reagents. We thank Bloomington *Drosophila* Stock Center and Vienna *Drosophila* Resource Center for fly stocks; Developmental Studies Hybridoma Bank for antibodies; and the fly community for sharing reagents. We thank Melissa A Visser for technical support. We also thank the Center for Integrated Microscopy (CFIM) at the University of Copenhagen for electron microscopy support.

## Author contributions

**Conceptualization:** Wei Qi Guinevere Sew, Maria Molano-Fernández, Héctor Herranz.

**Data curation:** Wei Qi Guinevere Sew, Maria Molano-Fernández, Héctor Herranz.

**Formal analysis:** Wei Qi Guinevere Sew, Maria Molano-Fernández, Artim Lange, Nahia Pérez de Ciriza.

**Funding acquisition:** Héctor Herranz.

**Investigation:** Wei Qi Guinevere Sew, Maria Molano-Fernández, Zhiquan Li, Artim Lange, Nahia Pérez de Ciriza, Lene Juel Rasmussen.

**Methodology:** Wei Qi Guinevere Sew, Maria Molano-Fernández, Zhiquan Li, Lene Juel Rasmussen, Héctor Herranz.

**Project administration:** Héctor Herranz.

**Supervision:** Héctor Herranz.

**Validation:** Maria Molano-Fernández, Héctor Herranz.

**Visualization:** Maria Molano-Fernández.

**Writing – original draft:** Héctor Herranz.

**Writing – review & editing:** Wei Qi Guinevere Sew, Maria Molano-Fernández, Zhiquan Li, Lene Juel Rasmussen, Héctor Herranz.

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
