## [Editor Report · Decision Letter 0]

21 Sep 2025

Dear Dr Herranz,

Thank you for submitting your manuscript via Review Commons entitled "Control of mitochondrial dynamics by dPGC1 limits Yorkie-induced oncogenic growth in Drosophila" for consideration as a Research Article by PLOS Biology.

Your manuscript and reviews have now been evaluated by the PLOS Biology editorial staff as well as by an academic editor with relevant expertise and I am writing to let you know that we would like to send your submission back to the original reviewers.

Once your full submission is complete, your paper will undergo a series of checks in preparation for peer review. After your manuscript has passed the checks it will be sent out for review. To provide the metadata for your submission, please Login to Editorial Manager (https://www.editorialmanager.com/pbiology) within two working days, i.e. by Sep 23 2025 11:59PM.

Kind regards,

Ines

--

Ines Alvarez-Garcia, PhD

Senior Editor

PLOS Biology

---

## [Decision Letter · Decision Letter 1]

17 Oct 2025

Dear Dr Herranz,

Thank you for your patience while we considered your revised manuscript from Review Commons entitled "Control of mitochondrial dynamics by dPGC1 limits Yorkie-induced oncogenic growth in Drosophila" for publication as a Research Article at PLOS Biology. This revised version of your manuscript has been evaluated by the PLOS Biology editors, an Academic Editor and two of the original reviewers.

Based on the reviews (attached below), we are likely to accept this manuscript for publication, provided you satisfactorily address the remaining minor points raised by Reviewer 3, and the data and other policy-related requests stated below my signature.

In addition, we would like you to consider a suggestion to improve the title:

“Control of mitochondrial dynamics by the metabolic regulator dPGC1 limits Yorkie-induced oncogenic growth in Drosophila"

We expect to receive your revised manuscript within two weeks.

*Published Peer Review History*

*Press*

Sincerely,

Ines

--

Ines Alvarez-Garcia, PhD

Senior Editor

PLOS Biology

Fig. 1B, E; Fig. 2E, I, M, N, Q; Fig. 3A; Fig. 4D, E; Fig. 5G; Fig. 6A, C, H, O; Fig. S1C; Fig. S2A-D; Fig. S3A-D; Fig. S4C, F; Fig. S5C; Fig. S6C; Fig. S7; Fig. S8C-E; Fig. S9; Fig. S10F; Fig. S11C, F; Fig. S13 and Fig. S14C

Reviewers' discussion

Rev. 1: Dane Wolf - note that this reviewer has signed the review.

The authors have addressed my comments adequately.

Rev. 3:

This revised manuscript represents a substantial step forward and now delivers a coherent and well-substantiated story. The authors convincingly demonstrate that the conserved co-activator dPGC1 functions as a context-dependent tumor suppressor in Drosophila, constraining Yki-driven oncogenic growth through control of mitochondrial dynamics and Cyclin E regulation. The combination of genetic precision, structural analysis (confocal and EM), and functional assays y+ ields a mechanistically grounded picture linking mitochondrial remodeling to cell-cycle control. The topic is timely and relevant across developmental biology, cancer metabolism, and Hippo signaling.

Major strengths

1. Rigorous revision and new experiments. The inclusion of loss-of-function alleles (dPGC1¹, Wts⁻/⁻), improved RNAi validation using act-GAL4, and expanded EM/confocal analyses greatly enhance robustness.

2. Mechanistic insight. The demonstration that Cyclin E dosage genetically modifies tumor growth (via CycE05306/+) establishes a functional link between mitochondrial dynamics and proliferation.

3. Context specificity. The observation that dPGC1 depletion does not potentiate EGFR or InR overgrowthconvincingly shows that this interaction is Yki-specific, addressing a key conceptual gap.

4. Integration with prior literature. The authors now situate their work alongside Nagaraj et al. 2012 and Spiegelman-lab studies, acknowledging precedent while delineating novelty.

5. Improved clarity and presentation. Figures now include quantified PH3, Mmp1, F-actin, and Western replicates; the narrative flow is logical and accessible.

Remaining issues for a final, publication-ready version

All points below are minor yet important refinements that will raise the manuscript to PLOS Biology's highest standard:

1. In the absence of an antibody, please state explicitly in Methods and Discussion that conclusions rest on transcript and genetic evidence, and discuss the limitation.

2. Supplement EM length data with an additional shape descriptor (aspect ratio or circularity) to distinguish elongation from swelling; even approximate metrics will strengthen claims.

3. The TMRE analysis is valuable; ensure inclusion of numerical quantification, n-values, and statistics in the figure legend, and briefly discuss how preserved MMP may indicate compensatory metabolic adjustments.

4. Expand the Discussion by speculating, succinctly, on plausible post-transcriptional mechanisms (e.g. altered degradation, ROS-mediated signaling) linking mitochondrial state to Cyclin E stabilization.

5. Provide a table detailing RNAi/shRNA lines, insertion sites, drivers, and efficienciesto aid reproducibility and transparency.

6. A careful proof for minor grammatical inconsistencies (articles, tense) will further polish the manuscript.

Overall assessment

The authors have been highly responsive and thorough, transforming an interesting initial study into a rigorous and conceptually compelling paper. The work establishes mitochondrial dynamics as an integral layer of Hippo pathway regulation and identifies dPGC1 as a safeguard against Yki-induced tumorigenesis - a finding with clear translational resonance.

Recommendation: Accept pending minor revision.

With these final adjustments, the manuscript will stand as a benchmark study for integrating metabolic control, organelle dynamics, and growth signaling in vivo.

---

## [Editor Report · Decision Letter 2]

11 Nov 2025

Dear Dr Herranz,

Thank you for the submission of your revised Research Article entitled "Control of mitochondrial dynamics by the metabolic regulator dPGC1 limits Yorkie-induced oncogenic growth in Drosophila" for publication in PLOS Biology. On behalf of my colleagues and the Academic Editor, Nic Tapon, I am delighted to let you know that we can in principle accept your manuscript for publication, provided you address any remaining formatting and reporting issues. These will be detailed in an email you should receive within 2-3 business days from our colleagues in the journal operations team; no action is required from you until then. Please note that we will not be able to formally accept your manuscript and schedule it for publication until you have completed any requested changes.

PRESS

Sincerely, 

Ines

--

Ines Alvarez-Garcia, PhD

Senior Editor

PLOS Biology
